# Spiking Graph Neural Network on Riemannian Manifolds

**Li Sun**[*]
North China Electric Power University
Beijing 102206, China
ccesunli@ncepu.edu.cn

**Zhenhao Huang**
North China Electric Power University
Beijing 102206, China
huangzhenhao@necpu.edu.cn

**Qiqi Wan**
North China Electric Power University
Beijing 102206, China
wanqiqi@ncepu.edu.cn

**Hao Peng**
Beihang University
Beijing 100191, China
penghao@buaa.edu.cn

**Philip S. Yu**
University of Illinois at Chicago
IL, USA
psyu@uic.edu

## Abstract

Graph neural networks (GNNs) have become the dominant solution for learning on graphs, the typical non-Euclidean structures. Conventional GNNs, constructed with the Artificial Neuron Network (ANN), have achieved impressive performance at the cost of high computation and energy consumption. In parallel, spiking GNNs with brain-like spiking neurons are drawing increasing research attention owing to the energy efficiency. So far, existing spiking GNNs consider graphs in Euclidean space, ignoring the structural geometry, and suffer from the high latency issue due to Back-Propagation-Through-Time (BPTT) with the surrogate gradient. In light of the aforementioned issues, *we are devoted to exploring spiking GNN on Riemannian manifolds*, and present a Manifold-valued Spiking GNN (`MSG`). In particular, we design a new spiking neuron on geodesically complete manifolds with the diffeomorphism, so that BPTT regarding the spikes is replaced by the proposed differentiation via manifold. Theoretically, we show that `MSG` approximates a solver of the manifold ordinary differential equation. Extensive experiments on common graphs show the proposed `MSG` achieves superior performance to previous spiking GNNs and energy efficiency to conventional GNNs.

## 1 Introduction

Graphs are the ubiquitous, non-Euclidean structures that describe the relationship among objects. Graph neural networks (GNNs), constructed with the floating-point Artificial Neuron Network (ANN), have achieved state-of-the-art accuracy for learning on graphs [1; 2; 3; 4]. However, they raise the concerns about computation and energy consumption, particularly when dealing with real-world graphs of considerable scale [5; 6]. In contrast, Spiking Neuron Networks (SNNs), inspired by the biological mechanism of brains, utilize neurons that communicate using sparse and discrete spikes, showcasing their superiority in energy efficiency [7; 8]. Attempting to bring the best of both worlds, **spiking GNNs** are drawing increasing research attention.

---

[*]Corresponding Author: Li Sun, ccesunli@ncepu.edu.cn

38th Conference on Neural Information Processing Systems (NeurIPS 2024).

In the literature of spiking GNNs, recent efforts have been made to design different architectures with spiking neurons, e.g., graph convolution [5], attention mechanism [9], variational autoencoder [10] and continuous GNN [11]. While achieving encouraging results, existing spiking GNNs still face several fundamental issues: **(1) Representation Space.** Spiking GNNs consider the graph in Euclidean space, ignoring the inherent geometry of graph structures. Unlike the Euclidean structures (e.g., pixel matrix and grid structures), graphs cannot be embedded in Euclidean space with bounded distortion [12]. Instead, *Riemannian manifolds* have been shown as the promising spaces to model graphs in recent years [3; 13; 4] (e.g., hyperbolic space, a type of Riemannian manifolds, is well aligned with the graphs dominated by hierarchical structures). However, none of the existing works study SNN on Riemannian manifolds, to the best of our knowledge. It is thus an interesting and urgent problem to consider how to endow the spiking GNN with a Riemannian manifold. **(2) Training Algorithm.** Training spiking GNN is challenging, since the spikes are non-differentiable. Existing studies consider the spiking GNN as a recurrent neural network and apply Backward-Passing-Through-Time (BPTT) with the surrogate gradient [5; 9; 10; 11]. They recurrently compute the backward gradient at each time step, and thus suffer from the high latency issue [14; 15; 16; 6] especially when the spike trains are long.

**Present work.** Deviating from previous spiking GNNs in Euclidean space, in this paper, we open a new direction to explore spiking GNNs on Riemannian manifolds, and propose a novel Manifold-valued Spiking GNN (MSG) sketched in Fig. 1. It is not realistic to place spike trains in a manifold such as hyperbolic or hyperspherical space, given the fact that spike trains cannot align with the defining domain. Instead, we design a Manifold Spiking Layer that conducts parallel forwarding of spike trains and manifold representations. Specifically, we first incorporate the structural information

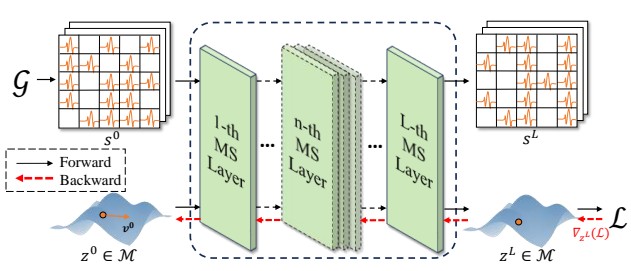

Figure 1: MSG conducts parallel forwarding and enables a new training algorithm alleviating the high latency issue.

into spike trains by graph convolution. Then, a new *manifold spiking neuron* is proposed to emit spike trains and relate them to manifold representations with *diffeomorphism*, where the spike train generates a momentum that forwards manifold representation along the geodesic. Instead of applying BPTT in spike domain, the proposed neuron provides us with an alternative of *Differentiation via Manifold* (*DvM*). (The red dashed line in Fig. 1.) Yet, differentiation in Riemannian manifold is nontrivial. We leverage the properties of pullback and derive the closed-form backward gradient (Theorem 4.1). *DvM* enables the recurrence-free gradient backpropagation, which no longer needs to perform recurrent computation of time steps as in BPTT. Theoretically, MSG is essentially related to manifold Ordinary Differential Equation (ODE). Each layer creates a *chart* of the manifold, and MSG approximates the dynamic chart solver [17] of manifold ODE (Theorem 5.2).

**Contributions.** Overall, the key contributions are summarized as follows: (1) To the best of our knowledge, we propose the first spiking neural network on Riemannian manifolds (MSG)[2], and show its connection to manifold ODE theoretically. (2) We design a new training algorithm of differentiation via manifold, which avoids the high latency of BPTT methods. (3) Extensive experiments show the superior effectiveness and energy efficiency of the proposed MSG.

## 2 Related Work

We briefly overview the ANN-based GNNs (i.e., the conventional, floating-point GNNs living in either Euclidean space or Riemannian manifolds) and SNN-based GNNs (i.e., spiking GNNs).

**ANN-based GNNs (Euclidean and Riemannian).** The majority of GNNs are built with floating-point ANN, conducting message passing on the graphs [18; 2; 19]. The Euclidean space has been the

---

[2]Codes are available at `https://github.com/ZhenhHuang/MSG`

workhorse for graph representation learning for decades, and the popular GCN [18], GAT [2] and SGC [19] are also designed in the Euclidean space. In recent years, Riemannian manifolds have emerged as an exciting alternative considering the geometry of graph structures [20; 21]. Among Riemannian manifolds, hyperbolic space is recognized for its alignment with the graphs of hierarchical structures, and a series of hyperbolic GNNs (e.g., HGNN [22], HGCN [3]) show superior performance to their Euclidean versions. Beyond hyperbolic space, hyperspherical space is well suited for cyclical structures [23], and recent studies further investigate the constant curvature spaces [13], product spaces [24; 25; 26; 27], quotient spaces [28], SPD manifolds [29; 30], etc. Riemannian manifolds achieve remarkable success in graph clustering [31; 32], structural learning [33], graph dynamics [34; 35; 36; 37], information diffusion [38] and graph generation [39; 40], but have rarely been touched yet in the SNN counterpart.

**Spiking Neural Networks (SNNs) & Spiking GNNs.** Mimicking the biological neural networks, SNNs [7; 8] utilize the spiking neuron to process spike trains, and offer the advantage of energy efficiency. Despite the wide application of SNN in computer vision [41; 42], SNNs are still at an early stage in the graph domain. The basic idea of spiking GNNs is adapting ANN-based GNNs to the SNN framework by substituting the activation functions with spiking neurons. Pioneering works study the graph convolution [43; 5], and efforts have also been made to the graph attention [9], variational graph autoencoder [10], graph differential equations [44], etc. SpikeGCL [6] is a recent endeavor to conduct graph contrastive learning with SNN. In parallel, spiking GNNs are extended to model the dynamic graphs [45; 46; 47]. We focus on the static graph in this work. In both dynamic and static cases, previous spiking GNNs are trained with the surrogate gradient, leading to high latency, and consider the graphs in the Euclidean space.

## 3 Preliminaries

Different from aforementioned spiking GNNs, we study the spiking GNN on Riemannian manifolds. Thus, we formally introduce the basic concepts of Riemannian geometry and SNN. Throughout this paper, the lowercase boldfaced $x$ and uppercase $\mathbf{X}$ denote vector and matrix, respectively. Important notations are summarized in Appendix A.

**Riemannian Geometry & Riemannian Manifold.** Riemannian geometry provides elegant framework to study structures and manifolds. A Riemannian manifold is described as a smooth and real manifold $\mathcal{M}$ endowed with a Riemannian metric. Each point $x$ in the manifold is associated with the *tangent space* $T_x\mathcal{M}$ that "looks Euclidean", and the Riemannian metric is given by the inner product in the tangent space, so that geometric properties (e.g, angle, length) can be defined. A *geodesic* between two points on the manifold is the smooth path connecting them with the minimal length. There exist three types of isotropic manifold, namely, the *Constant Curvature Space* (CCS): hyperbolic space $\mathbb{H}$, hyperspherical space $\mathbb{S}$ and the special case of Euclidean space with "flat" geometry $\mathbb{E}$.

**Graph & Riemannian Graph Representation Learning.** A graph $\mathcal{G} = (\mathcal{V}, \mathcal{E}, \mathbf{F}, \mathbf{A})$ is defined on the node set $\mathcal{V}$ and edge set $\mathcal{E} \subset \mathcal{V} \times \mathcal{V}$, and $\mathbf{A} \in \mathbb{R}^{|\mathcal{V}| \times |\mathcal{V}|}$ is the adjacency matrix describing the structure information. Each node $v_i$ is associated with a feature $\boldsymbol{f}_i$, and node features are summarized in $\mathbf{F} \in \mathbb{R}^{|\mathcal{V}| \times d}$. In this paper, we resolve the problem of *Riemannian Graph Representation Learning* with SNN. Specifically, we seek a graph encoder $\mathcal{F}_\theta : v \mapsto z$ where $z \in \mathcal{M}$ is a point on the manifold, instead of Euclidean space, and $\mathcal{F}_\theta$ is defined with an energy-efficient SNN.

**Spiking Neural Network.** SNNs are constructed by *spiking neurons* that communicate with each other by spike trains. Concretely, a spiking neuron is conceptualized as "a capacitor of the membrane potential", and processes the spike trains by the following 3 phases [48]. First, the incoming current $I[t]$ is accumulated in the capacitor, leading to the potential buildup (*integrate*). When the membrane potential $V[t]$ reaches or exceeds a specific threshold $V_{th}$, the neuron emits a spike (*fire*). After that, the membrane potential is reset to the resting potential $V_{reset}$ (*reset*). There are two popular spiking neurons: IF model and LIF model [49]. In particular, the three phases of IF model are formalized as

$$\text{Integrate}: \quad V[t] = g(V[t-1], I[t]) = V[t-1] + I[t] \tag{1}$$

$$\text{Fire}: \quad S[t] = H(V[t] - V_{th}) \tag{2}$$

$$\text{Reset}: \quad V[t] = \begin{cases} (1 - S[t])V[t] + S[t]V_{rest}, & \textit{Fixed-reset}, \\ V[t] - V_{th}S[t], & \textit{Subtraction-reset}. \end{cases} \tag{3}$$

where the incoming current $I[t]$ is related to the input spike train, and $V_{reset}$ is lower than $V_{th}$. $t$ denotes the time index of the spike. The Heaviside function $H(\cdot)$ is non-differentiable, $H(x) = 1$ if $x \geq 0$, and 0 otherwise. There are two options for reset, and fixed-reset is adopted in this paper. Overall, an IF model is given as $S[t] = \text{IFModel}(I[t])$, and the only difference between IF model and LIF model lies in the definition of $g(\cdot)$ in Eq. (1). In this paper, we are interested in designing a new spiking neuron on Riemannian manifold.

## 4 Methodology: Manifold-valued Spiking GNN

In this section, we present a simple yet effective Manifold-valued Spiking GNN (MSG), which can be applied to any geodesically complete manifolds, e.g., the Constant Curvature Space (CCS), including hyperbolic space and hyperspherical space, or the product of CCS. In particular, we design a spiking neuron on Riemannian manifolds (named as *Manifold Spiking Neuron*) that allows for the *differentiation via manifold*. It provides a new perspective of training spiking GNN, so that we avoid the high latency of typical backward-passing-through-time (BPTT) training.

### 4.1 Manifold Spiking Layer

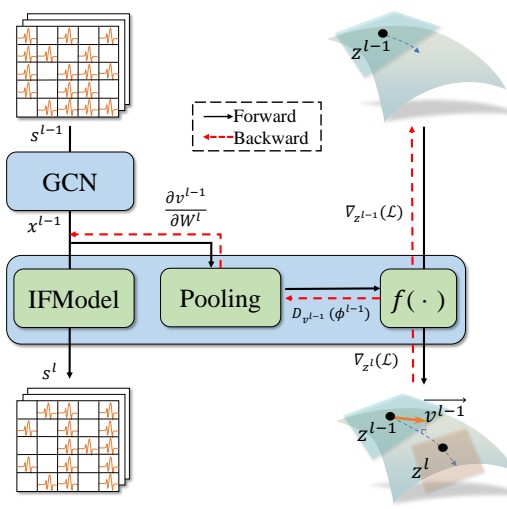

Figure 2: Manifold Spiking Layer. It conducts parallel forwarding of spike trains and manifold representations, and creates an alternative backward pass (red dashed line). The backward gradient with $\frac{\partial \mathbf{v}^{l-1}}{\partial \mathbf{W}^l}$, $D_{\mathbf{v}^{l-1}}\phi^{l-1}$ and $\nabla_{\mathbf{z}^l}\mathcal{L}$ will be introduced in Sec. 4.2.

We elaborate on the sole building block of the proposed model — Manifold Spiking Layer. Note that, the spike train or spiking representation in existing spiking GNNs [43; 5; 9; 10; 6; 45; 46; 47] cannot align with the defining domain of Riemannian manifolds (e.g., hyperbolic space and hyperspherical space), thus posing a fundamental challenge. Our solution is to generate node representation on the manifold (referred to as manifold representation) in parallel, and leverage the notion of Diffeomorphism to create the alignment between the two domains. We formulate the *parallel forwarding* of spike trains and manifold representations as follows.

**Unified Formulation.** The forward pass of the spiking layer consists of a graph convolution and one proposed manifold spiking neuron. Without loss of generality, for each node $v_i \in \mathcal{G}$, the $l$−th spiking layer is formulated as follows,

$$\mathbf{x}_i^{l-1}[t] = \text{GCN}(\mathbf{s}_i^{l-1}[t]; \mathbf{W}^l), \qquad (4)$$

$$[\mathbf{s}_i^l, \mathbf{z}_i^l] = \text{MSNeuron}(\mathbf{x}_i^{(l-1)}, \mathbf{z}_i^{(l-1)}), \quad (5)$$

where $\mathbf{x}$ is the incoming current to generate spike trains. $\mathbf{s}$ and $\mathbf{z}$ denote the spike trains and manifold representation, respectively. $\text{GCN}(\cdot)$ is a GNN in the Euclidean space, and $\mathbf{W}^l$ is the learnable parameter in the layer. Different from the neuron of previous spiking GNNs, we design *a novel manifold spiking neuron* (MSNeuron) as shown in Fig. 2. It emits spike trains and relates them to manifold representations simultaneously, which is formulated as follows,

$$\mathbf{s}_i^l = \text{IFModel}(\{\mathbf{x}_i^{l-1}[t]\}_{t=1,\dots,T}) \qquad (6)$$

$$\mathbf{v}_i^{l-1} = \text{Pooling}(\mathbf{x}_i^{l-1}[t]) \qquad (7)$$

$$\mathbf{z}_i^l = f(\mathbf{z}_i^{l-1}, \epsilon\mathbf{v}_i^{l-1}) \in \mathcal{M} \qquad (8)$$

where $t$ is the time step of spike trains. The IF model can be replaced by LIF model, and we utilize IF model by default for simplicity. Pooling is defined as the average pooling of the current $\mathbf{x}$ over $t$, and $\mathbf{v}$ is given as the Euclidean vector. $f$ denotes the diffeomorphism to the Riemannian manifold in which $\epsilon$ is the step size.

**Incorporating structural information.** We inject the structural information when the received spikes transform into the incoming current of the neuron. A GNN is leveraged to define the current, conducting message-passing over the graph. Each node's representation is derived recursively by neighborhood aggregation [18; 2; 1]. Accordingly, GCN in the proposed neuron is given as follows,

$$\text{GCN}(\mathbf{s}_i^{l-1}[t]; \mathbf{W}^l) = \text{combine}(\mathbf{s}_i^{l-1}[t], \text{aggregate}(\{\mathbf{s}_j[t] : j \in \Omega_i\}; \mathbf{W}^l)), \quad (9)$$

where the neighborhood $\Omega_i$ is the set of immediate neighbors centering at node $v_i$. The aggregate function aggregates the messages from neighborhood $\Omega_i$, where we create the message of a node by $\mathbf{W}^l \mathbf{s}_j[t]$. combine$(\cdot)$ denotes the combination of the center node's message and aggregated message. We utilize GCN [18] as the backbone to define aggregate and combine.

**Diffeomorphism between manifolds.** In the proposed neuron, we bridge the spikes and manifold representation with the notion of *diffeomorphism* in differential geometry. A diffeomorphism connects two smooth manifolds, saying $\mathcal{M}$ and $\mathcal{N}$. Formally, a map $f : \mathcal{M} \to \mathcal{N}$ is a diffeomorphism between $\mathcal{M}$ and $\mathcal{N}$ if the smooth $f$ is bijective and its inverse $f^{-1}$ is also smooth.

Recall that the tangent space is locally Euclidean. We propose to place the Euclidean $\mathbf{v}$, a representation of the spikes, in the tangent space $T_{\mathbf{z}}\mathcal{M}$ of the point $\mathbf{z}$. In MSG, we choose the *exponential map* to act as the diffeomorphism between the tangent space and manifold. With a step size $\epsilon$, we have

$$f(\mathbf{z}_i^{(l-1)}, \epsilon \mathbf{v}_i^{(l-1)}) = \text{Exp}_{\mathbf{z}_i^{(l-1)}}(\epsilon \mathbf{v}_i^{(l-1)}) \in \mathcal{M} \quad (10)$$

Concretely, given $\mathbf{z} \in \mathcal{M}$ and $\mathbf{v} \in T_{\mathbf{z}}\mathcal{M}$, the exponential map[3] of $\mathbf{v}$ at point $\mathbf{z}$, $\text{Exp}_{\mathbf{z}}(\mathbf{v}) : T_{\mathbf{z}}\mathcal{M} \to \mathcal{M}$, maps tangent vector $\mathbf{v}$ onto the manifold $\mathcal{M}$. The map pushes $\mathbf{z}$ along the *geodesic* $\gamma_{\mathbf{z},\mathbf{v}}(t) : [0, 1] \to \mathcal{M}$ starting at $\gamma_{\mathbf{z},\mathbf{v}}(0) = \mathbf{z}$ and ending at $\mathbf{y} = \gamma_{\mathbf{z},\mathbf{v}}(1)$. $\dot{\gamma}_{\mathbf{z},\mathbf{v}}(t)$ denotes the velocity of $\gamma_{\mathbf{z},\mathbf{v}}(t)$, and the direction of geodesic at the beginning is given as $\dot{\gamma}_{\mathbf{z},\mathbf{v}}(0) = \mathbf{v}$. That is, the tangent vector $\mathbf{v}$, derived from the spikes, pushes the manifold representation along the geodesic via the exponential map. The advantage of our choice is that we are able to define the diffeomorphism in arbitrary geodesically complete manifold (detailed in Appendix D).

Note that, our idea is inherently different from the exponential/logarithmic based Riemannian GNNs [3; 4; 22], which leverage the tangent space of the origin for neighborhood aggregation. In contrast, we consider the successive process over the tangent spaces of manifold representations, which will be further studied in Sec. 5.

**Model Initialization** In MSG, we need to simultaneously initialize the spiking input $\mathbf{S}^0$ and manifold representation $\mathbf{Z}^0$, which is a collection of points on the given manifold. Given a graph $\mathcal{G}(\mathcal{V}, \mathcal{E}, \mathbf{F}, \mathbf{A})$, the node features are first encoded by one graph convolution layer $\mathbf{H} = \text{GCN}(\mathbf{A}, \mathbf{F}; \mathbf{W}^0)$, and we generate $T$ copies of the node encodings $\mathbf{H}$, where $T$ is the number of time steps in spike trains. Then, we complete model initialization with the proposed manifold neuron $[\mathbf{S}^0, \mathbf{Z}^0] = \text{MSNeuron}(\mathbf{H}, \mathbf{O})$, where the encoding $\mathbf{H}$ is regarded as the incoming current that charges the neuron in each time step. $\mathbf{O}$ consists of the original points of the manifold, e.g., in the sphere model of hyperspherical space, the original point is given as the south pole $\mathbf{o} = [-1, 0, ..., 0]^\top$ and $\mathbf{O} = [\mathbf{o}^\top, ..., \mathbf{o}^\top]^\top$. Note that, the exponential map in the proposed neuron guarantees that $\mathbf{Z}^0$ lives in the manifold.

### 4.2 Learning Approach: Differentiation via Manifold

Optimizing SNNs is challenging, as the Heaviside step function is non-differentiable. In the literature, existing spiking GNNs typically regard SNN as the recurrent neural network, and leverage backward-passing-through-time (BPTT) to train the model [50; 51; 52]. Concretely, given a real loss function $\mathcal{L}$, the gradient backpropagation conducts **Differentiation via Spikes** (*DvS*) **s** as follows,

$$\nabla_{\mathbf{W}^l} \mathcal{L} = \sum_t [\frac{\partial \mathbf{s}^l[t]}{\partial \mathbf{W}^l}]^* \nabla_{\mathbf{s}^l[t]} \mathcal{L}, \quad (11)$$

where $\mathbf{W}$ is the parameter, and $t$ denotes the time step. The surrogate gradient [51] is required for $D_{\mathbf{W}^l} \mathbf{s}^l[t]$, where Heaviside step function is replaced by a differentiable surrogate, e.g., sigmoid function. The differentiation via spikes presents high latency in the backward pass [14; 15; 16], as it needs to recur all the time steps in BPTT. We notice that, in the computer vision domain, the sub-gradient method [15] is proposed to address such issues in Euclidean space. However, it cannot be generalized to the Riemannian manifold since the linearity does not hold in Riemannian geometry.

---

[3]The inverse map from $\mathcal{M}$ to $T_{\mathbf{z}}\mathcal{M}$ is the logarithmic map.

In `MSG`, we decouple the forward pass and backward pass, and propose **Differentiation via Manifold** (*DvM*) to avoid the high latency in differentiation via spikes. The overall procedure of training `MSG` by the proposed learning approach is summarized in Algorithm 1. Thanks to the parallel forwarding of spikes and manifold representation, the proposed neuron provides us with an alternative of studying $\nabla_{\mathbf{W}^l}\mathcal{L}$ through the forwarding pass on the manifold (i.e., differentiation via manifold). Nevertheless, *it is nontrivial and it requires to derive the pullback between different dual spaces.*

**Pushforward, Pullback and Dual Space.** We first introduce the differentiation in Riemannian geometry which is essentially different from that in Euclidean space. In Riemannian geometry, a *pushforward* refers to a derivative of a map connecting two manifolds $\mathcal{M}$ and $\mathcal{N}$. Concretely, given $f : \mathcal{M} \rightarrow \mathcal{N}$ and a point $\mathbf{z} \in \mathcal{M}$, the pushforward $D_{\mathbf{z}}f$ maps a tangent vector $\mathbf{v} \in T_{\mathbf{z}}\mathcal{M}$ to the tangent vector $D_{\mathbf{z}}f(\mathbf{v}) \in T_{f(\mathbf{z})}\mathcal{N}$. On the notation, for a manifold-valued function $f(\mathbf{z}) = \mathbf{p} \in \mathcal{N}$, $\partial\mathbf{p}/\partial\mathbf{z}$ is equivalent to $D_{\mathbf{z}}f$. For a scalar function $f$, $D_{\mathbf{z}}f$ is interchangeable with $\nabla_{\mathbf{z}}f$.

---

**Algorithm 1** Training `MSG` by the proposed Differentiation via Manifold

**Input:** Graph $\mathcal{G}(\mathcal{V}, \mathcal{E}, \mathbf{F}, \mathbf{A})$, Manifold $\mathcal{M}$, Loss function over the manifold $\mathcal{L}(\cdot)$, Number of spiking layers $L$, Original points $\mathbf{O}$.
**Output:** Parameters $\{\mathbf{W}^l\}_{l=0,\cdots,L}$
1: **while** not converging **do**
2:     ▷ *forward pass*
3:     Input current $\mathbf{X}^0 = \text{GCN}(\mathbf{A}, \mathbf{F}; \mathbf{W}^0)$;
4:     Initialize $[\mathbf{S}^0, \mathbf{Z}^0] = \text{MSNeuron}(\mathbf{X}^0, \mathbf{O})$;
5:     **for** each spiking layer $l = 1$ to $L$ **do**
6:         $\mathbf{X}^{(l-1)} = \text{GCN}(\mathbf{A}, \mathbf{S}^{(l-1)}; \mathbf{W}^l)$;
7:         $[\mathbf{S}^l, \mathbf{Z}^l] = \text{MSNeuron}(\mathbf{X}^{(l-1)}, \mathbf{Z}^{(l-1)})$;
8:     **end for**
9:     ▷ *backward pass*
10:    Compute $\nabla_{\mathbf{z}^L}\mathcal{L}$ from $\mathcal{L}(\mathbf{Z}^L)$.
11:    **for** layer $l = L - 1$ to $1$ **do**
12:        Compute $D_{\mathbf{z}^l}\psi^l, D_{\mathbf{v}^{l-1}}\phi^{l-1}, \frac{\partial\mathbf{v}^{l-1}}{\partial\mathbf{W}^l}$.
13:        Compute $\nabla_{\mathbf{z}^l}\mathcal{L}, \nabla_{\mathbf{W}^l}\mathcal{L}$ as Eq. 12.
14:        Update $\mathbf{W}^l$.
15:    **end for**
16: **end while**

---

In the proposed `MSG`, we consider a scalar loss function on the manifold $\mathcal{L} : \mathcal{M} \rightarrow \mathbb{R}$. The pushforward $D_{\mathbf{z}}\mathcal{L}$ at point $\mathbf{z} \in \mathcal{M}$ maps tangent vector $\mathbf{v} \in T_{\mathbf{z}}\mathcal{M}$ to a scalar value and, correspondingly, $D_{\mathbf{z}}\mathcal{L}$ belongs to the *dual space* of the tangent space $T_{\mathbf{z}}^*\mathcal{M}$, which is a vector space consisting all linear functional $F : T_{\mathbf{z}}\mathcal{M} \rightarrow \mathbb{R}$. As the tangent spaces at different points of the manifold are different, it requires a *pullback* that maps the dual space $T_{\mathbf{z}^l}^*\mathcal{M}$ to the dual space $T_{\mathbf{z}^{(l+1)}}^*\mathcal{M}$.

We derive the backward gradient with properties of differential $1-$form (Lemma B.1), communication (Lemma B.2), and pullback of a sum and a product (Lemma B.3) detailed in Appendix B.1.

**Theorem 4.1** (Backward Gradient). *Let $\mathcal{L}$ be the scalar-valued function, and $\mathbf{z}^l$ is the output of l-th layer with parameter $\mathbf{W}^l$, which is delivered by tangent vector $\mathbf{v}^l$. Then, the gradient of function $\mathcal{L}$ w.r.t $\mathbf{W}^l$ is given as follows:*

$$\nabla_{\mathbf{W}^l}\mathcal{L} = [\frac{\partial\mathbf{v}^{l-1}}{\partial\mathbf{W}^l}]^*[D_{\mathbf{v}^{l-1}}\phi^{l-1}]^*\nabla_{\mathbf{z}^l}\mathcal{L}, \quad \nabla_{\mathbf{z}^l}\mathcal{L} = [D_{\mathbf{z}^l}\psi^l]^*\nabla_{\mathbf{z}^{l+1}}\mathcal{L}, \quad (12)$$

*where $\phi^{l-1}(\cdot) = \text{Exp}_{\mathbf{z}^{l-1}}(\cdot)$, $\psi^l(\cdot) = \text{Exp}_{(\cdot)}(\mathbf{v}^l)$, and $[\cdot]^*$ means the matrix form of pullback.*

The detailed proof is given in Appendix B.1, and we derive the two Jacobian matrices $D_{\mathbf{v}^{l-1}}\phi^{l-1}$ and $D_{\mathbf{z}^l}\psi^l$ in Appendix C. There are three key advantages of the proposed *DvM*. First, every term in Equation (12) is **differentiable**, and thereby the surrogate gradient is no longer needed. Second, *DvM* enables the **recurrence-free** backward pass alleviating the high latency training. We specify that both *DvM* and the previous *DvS* recurrently compute every time step in the forward pass, and the difference lies in the backward pass. In particular, we only conduct recurrence-free gradient backpropagation layer by layer, while the previous *DvS* recurs every time step of each layer in BPTT. In addition to the differentiable and recurrence-free properties, *DvM* does not suffer from gradient vanishing/explosion, and the empirical evidence is provided in Appendix F.

## 5 Theory: MSG as Neural ODE Solver

Next, we demonstrate the theoretical aspects of our model that *MSG approximates a solver of manifold Ordinary Differential Equations (ODEs).*

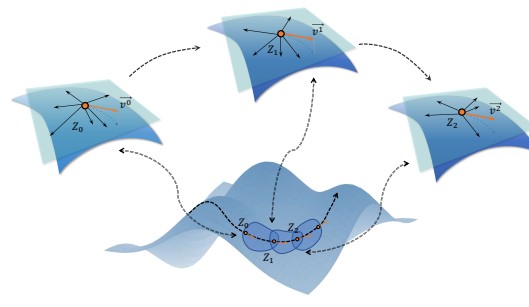

Figure 3: Charts given by the logarithmic map.

We leverage the notion of **chart** to study the relationship between MSG and manifold ODE. A *manifold ODE* defined as

$$\frac{d\mathbf{z}(t)}{dt} = u(\mathbf{z}(t), t), \quad \mathbf{z}(t) \in \mathcal{M}, \quad (13)$$

describes a *vector field* $u$ that maps a smooth path $\mathbf{z}(t) : [0, 1] \to \mathcal{M}$ to the tangent bundle $T\mathcal{M}$ [4]. In other words, the vector field $u$ assigns each point $\mathbf{z}(t) \in \mathcal{M}$ to a tangent vector $u(\mathbf{z}(t), t) \in T_{\mathbf{z}(t)}\mathcal{M}$. A *chart* at $\mathbf{z}$, denoted as $(U_{\mathbf{z}}, \phi_{\mathbf{z}})$, is a smooth bijection $\phi_{\mathbf{z}}$ between $\mathbf{z}$'s neighborhood $U_{\mathbf{z}} \subset \mathcal{M}$ and a subspace of Euclidean space. Thus, the chart relates Eq. (13) to ODE in Euclidean space [17]. Specifically, if $\mathbf{y}(t) : [\tau, \tau + \epsilon] \to \mathbb{R}^n$ is the solution of

$$\frac{d\mathbf{y}(t)}{dt} = (D_{\phi_i^{-1}(\mathbf{y}(t))}\phi_i)u(\phi_i^{-1}(\mathbf{y}(t)), t), \quad (14)$$

then $\mathbf{y}(t) = \phi_i(\mathbf{z}(t))$ is a valid solution of Eq. (13) on $t \in [\tau, \tau + \epsilon]$.

**Definition 5.1** (Dynamic Chart Solver [17])**.** *The manifold ODE in Eq. (13) with initial condition $\mathbf{z}(0) = \mathbf{z}$ can be solved with a finite collection of successive charts $\{(U_i, \phi_i)\}_{i=1,...,L}$. If $\mathrm{ode}_i$ is the numerical solver to Euclidean ODE corresponding to the $i$-th chart, $\mathbf{y}(t) = \mathrm{ode}_i(t)$ on $[\tau_i, \tau_i + \epsilon_i]$, then $\mathbf{z}(t)$ in Eq. (13) is given as*

$$(\phi_L^{-1} \circ \mathrm{ode}_L \circ(\phi_L \circ \phi_{L-1}^{-1}) \circ ... \circ (\phi_2 \circ \phi_1^{-1}) \circ \mathrm{ode}_1 \circ \phi_1)(t). \quad (15)$$

That is, a manifold ODE can be solved in Euclidean subspaces given by a series of successive charts.

In MSG, we consider the charts given by the logarithmic map as illustrated in 3, and we prove that MSG approximates a dynamic chart solver of manifold ODE (Theorem 5.2).

**Theorem 5.2** (MSG as Dynamic Chart Solver)**.** *If $\mathbf{y}(t) : [\tau, \tau + \epsilon] \to \mathbb{R}^n$ is the solution of*

$$\frac{d\mathbf{y}(t)}{dt} = (D_{\mathrm{Exp}_{\mathbf{z}}(\mathbf{y(t)})} \mathrm{Log}_{\mathbf{z}})u(\mathrm{Exp}_{\mathbf{z}}(\mathbf{y}(t)), t), \quad (16)$$

*then $\mathbf{z}(t) = \mathrm{Exp}_{\mathbf{z}}(\mathbf{y}(t))$ is a valid solution to the manifold ODE of Eq. (13) on $t \in [\tau, \tau + \epsilon]$, where $\mathbf{z} = \mathbf{z}(\tau)$. If $\mathbf{y}(t)$ is given by the first-order approximation with the $\epsilon$ small enough,*

$$\mathbf{y}(\tau + \epsilon) = \epsilon \cdot (D_{\mathbf{z}} \mathrm{Log}_{\mathbf{z}})u(\mathbf{z}(\tau), \tau), \quad (17)$$

*then the update process of Eqs. (4) and (8) in MSG is equivalent to Dynamic Chart Solver in Eq. (15).*

*Proof.* The proof utilizes some facts of Riemannian manifolds and is detailed in Appendix B.2. □

In other words, in MSG, the transformation of manifold input and output is described as some manifold ODE, whose vector field is governed by a spiking-related neural network in the tangent bundle. To solve the manifold ODE, MSG leverages the Dynamic Chart Solver (Definition 5.1). Specifically, each manifold spiking layer corresponds to a chart, and thus the number of spiking layers equals to the number of charts. Each layer solves the ODE of a smooth path $\mathbf{y}(t) : [\tau, \tau + \epsilon] \to \mathbb{R}^n$ in the tangent space that centered at the manifold layer input. With the first-order approximation in Theorem 5.2, given a step size $\epsilon$, the endpoint $\mathbf{y}(\tau + \epsilon)$ of the path is parameterized by a GNN related to the spikes. Layer-by-layer forwarding solves the manifold ODE from the current chart to the successive chart. Consequently, the manifold output of MSG approximates the solution to the manifold ODE.

We notice that a recent work [11] connects spiking GNN to an ODE in Euclidean space. In contrast, the proposed MSG is essentially related to the manifold ODE.

The **Appendix** contains the proofs, the derivation of Jacobian, necessary facts on Riemannian geometry (i.e., Lorentz/Sphere model, stereographic projection and $\kappa$-stereographic model, and Cartesian product and product space), empirical details and additional results.

---

[4] The tangent bundle $T\mathcal{M}$ is the disjoint union of all the tangent spaces of the manifold $T\mathcal{M} = \bigsqcup_{\mathbf{z} \in \mathcal{M}} T_{\mathbf{z}}\mathcal{M}$.

Table 1: Node Classification (NC) in terms of classification accuracy (%) and Link Prediction in terms of AUC (%) on Computers, Photo, CS and Physics datasets. The best results are **boldfaced**, and the runner-ups are underlined. The standard derivations are given in the subscripts.

| | | Computers | | Photo | | CS | | Physics | |
|---|---|---|---|---|---|---|---|---|---|
| | | NC | LP | NC | LP | NC | LP | NC | LP |
| ANN-E | GCN [18] | $83.55_{\pm0.71}$ | $92.07_{\pm0.40}$ | $86.01_{\pm0.20}$ | $88.84_{\pm0.39}$ | $91.68_{\pm0.84}$ | $93.68_{\pm0.84}$ | $95.03_{\pm0.19}$ | $93.46_{\pm0.39}$ |
| | GAT [2] | $86.82_{\pm0.04}$ | $91.91_{\pm1.08}$ | $86.68_{\pm1.32}$ | $88.45_{\pm0.07}$ | $91.74_{\pm0.22}$ | $94.06_{\pm0.70}$ | $95.11_{\pm0.29}$ | $93.44_{\pm0.70}$ |
| | SGC [19] | $82.17_{\pm1.25}$ | $90.46_{\pm0.80}$ | $87.91_{\pm0.65}$ | $89.84_{\pm0.40}$ | $92.09_{\pm0.05}$ | $\mathbf{95.94_{\pm0.43}}$ | $94.77_{\pm0.32}$ | $95.93_{\pm0.70}$ |
| | SAGE [1] | $81.69_{\pm0.86}$ | $90.56_{\pm0.48}$ | $89.41_{\pm1.28}$ | $89.86_{\pm0.90}$ | $\mathbf{92.71_{\pm0.73}}$ | $95.22_{\pm0.14}$ | $95.62_{\pm0.26}$ | $95.75_{\pm0.37}$ |
| ANN-R | HGCN [3] | $88.71_{\pm0.24}$ | $96.88_{\pm0.53}$ | $89.18_{\pm0.50}$ | $94.54_{\pm0.20}$ | $90.72_{\pm0.16}$ | $93.02_{\pm0.26}$ | $94.46_{\pm0.20}$ | $94.10_{\pm0.64}$ |
| | $\kappa$-GCN [13] | $89.20_{\pm0.50}$ | $95.30_{\pm0.20}$ | $92.22_{\pm0.62}$ | $94.89_{\pm0.15}$ | $91.98_{\pm0.15}$ | $94.86_{\pm0.18}$ | $95.85_{\pm0.20}$ | $94.58_{\pm0.22}$ |
| | $\mathcal{Q}$-GCN [4] | $85.94_{\pm0.93}$ | $\mathbf{96.98_{\pm0.05}}$ | $92.50_{\pm0.95}$ | $97.47_{\pm0.03}$ | $91.18_{\pm0.28}$ | $93.39_{\pm0.20}$ | $94.84_{\pm0.25}$ | OOM |
| | HyboNet [54] | $86.29_{\pm2.30}$ | $96.80_{\pm0.05}$ | $92.67_{\pm0.09}$ | $\mathbf{97.70_{\pm0.07}}$ | $92.34_{\pm0.03}$ | $95.65_{\pm0.26}$ | $95.56_{\pm0.18}$ | $\mathbf{98.46_{\pm0.49}}$ |
| SNN-E | SpikeNet [45] | $88.00_{\pm0.70}$ | - | $92.90_{\pm0.10}$ | - | $92.15_{\pm0.18}$ | - | $92.66_{\pm0.30}$ | - |
| | SpikeGCN [5] | $86.90_{\pm0.30}$ | $91.12_{\pm1.79}$ | $92.60_{\pm0.70}$ | $93.84_{\pm0.03}$ | $90.86_{\pm0.11}$ | $95.07_{\pm1.22}$ | $94.53_{\pm0.18}$ | $92.88_{\pm0.80}$ |
| | SpikeGCL [6] | $89.04_{\pm0.08}$ | $92.72_{\pm0.03}$ | $92.50_{\pm0.17}$ | $95.58_{\pm0.11}$ | $91.77_{\pm0.11}$ | $95.13_{\pm0.24}$ | $95.21_{\pm0.10}$ | $94.15_{\pm0.29}$ |
| | SpikeGT [55] | $81.00_{\pm1.06}$ | - | $90.66_{\pm0.38}$ | - | $91.86_{\pm0.61}$ | - | $94.38_{\pm1.57}$ | - |
| | MSG (Ours) | $\mathbf{89.27_{\pm0.19}}$ | $94.65_{\pm0.73}$ | $\mathbf{93.11_{\pm0.11}}$ | $96.75_{\pm0.18}$ | $92.65_{\pm0.04}$ | $95.19_{\pm0.15}$ | $\mathbf{95.93_{\pm0.07}}$ | $93.43_{\pm0.16}$ |

# 6 Experiments

We conduct extensive experiments with 12 strong baselines to evaluate the proposed MSG in terms of (1) the representation effectiveness, (2) the energy efficiency, and (3) the advantages of the proposed components. Additional results are presented in Appendix F.

## 6.1 Experimental Setups

**Datasets.** Our experiments are conducted on 4 commonly used benchmark datasets including two popular co-purchase graphs: *Computers* and *Photo*[53], and two co-author graphs: *CS* and *Physics* [53]. Datasets are detailed in Appendix E.

**Baselines.** We compare the proposed MSG with 12 strong baselines of three categories: (1) *ANN-based Euclidean GNNs*: the popular GCN [18], GAT [2], GraphSAGE [1] and SGC [19], (2) *ANN-based Riemannian GNNs*: HGCN [3] and HyboNet [54] of hyperbolic spaces, $\kappa-$GCN [13] of the constant curvature space, and the recent $Q-$GCN [4] of the quotient space, (3) *The previous Euclidean Spiking GNNs*: SpikeNet [45], SpikeGCN [5], SipkeGraphormer [55] (termed as SpikeGT for short) and the recent SpikeGCL [6]. Note that, we focus on the graph representation learning on static graphs, and thereby graph models for the dynamic ones are out of the scope of this paper. SpikeNet [45] was originally designed for dynamic graphs, and we utilize its variant for static graphs according to [6]. So far, spiking GNN has not yet been connected to Riemannian manifolds, and we are devoted to bridging this gap.

**Evaluation Protocol.** All models are evaluated by node classification and link prediction tasks. The evaluation metrics of node classification is classification accuracy; we employ the popular Area Under Curve (AUC) for link prediction. The hyperparameter setting is the same as the original papers. We perform 10 independent runs for each case, and report the mean with standard derivations. Experiments are conducted on the hardware of NVIDIA GeForce RTX 4090 GPU 24GB memory, and AMD EPYC 9654 CPU with 96-Core Processor. Our model is built upon GeoOpt [56], SpikingJelly [56] and PyTorch [57].

**Model Instantiation & Configuration.** Note that, the proposed MSG applies to any Constant Curvature Space (CCS) or the product of CCS. We instantiate MSG in the Lorentz model of hyperbolic space by default (whose Riemannian metric, exponential map, and the derived Jacobian is given in Appendix C), and study the impact of representation space in the Ablation Study. The dimension of the representation space is set as 32. The manifold spiking neuron is based on the IF model [49] by default, and it is ready to switch to the LIF model [49] whose results are given in Appendix F. The time steps $T$ for neurons is set to 5 or 15. The step size $\epsilon$ in Eq. 8 is set to 0.1. The hyperparameters are tuned with grid search, in which the learning rate is $\{0.01, 0.003\}$ for node classification and $\{0.003, 0.001\}$ for link prediction, and the dropout rate is in $\{0.1, 0.3, 0.5\}$. We provide the source code of MSG at the anonymous link `https://github.com/ZhenhHuang/MSG`.

## 6.2 Results & Discussion

**Effectiveness.** We evaluate the effectiveness of `MSG` in both node classification and link prediction tasks. Specifically, for node classification, we cannot directly feed the manifold representations of Riemannian baselines to a softmax layer with Euclidean measure. We bridge the manifold representation and Euclidean softmax with the logarithmic map of respective manifold. For link prediction, we utilize the generalized sigmoid for all the baselines, i.e., the Fermi-Dirac decoder [3] in which the distance function is defined under the respective geometry. In the proposed MSG, the model inference does not need the expensive successive exponential maps, and only limited float-point operations (i.e., addition) are involved. Accordingly, we leverage the tangent vectors for the downstream tasks. The performance of both learning tasks on Computer, Photo, CS and Physics datasets are collected in Table 1. Note that, SpikeNet and SpikeGT cannot do link prediction, since they are designed for node classification and do not offer spiking representation. *The proposed* `MSG` *consistently achieves the best results among SNN-based models.* In addition, `MSG` generally outperforms the best ANN-based baselines in node classification, and has competitive results to the recent ANN-based Riemannian baselines in link prediction.

**Ablation Study.** Here, we examine the impact of representation space and the effectiveness of the proposed *Differentiation via Manifold (DvM)*. For the former goal, we instantiate 6 geometric variants of `MSG` in hyperbolic space $\mathbb{H}^{32}$, hyperspherical space $\mathbb{S}^{32}$, Euclidean space $\mathbb{E}^{32}$ and the products of $\mathbb{H}^{16} \times \mathbb{H}^{16}$, $\mathbb{H}^{16} \times \mathbb{S}^{16}$ and $\mathbb{S}^{16} \times \mathbb{S}^{16}$. The superscript denotes the dimension

Table 2: Ablation study of geometric variants. Results of node classification in terms of ACC (%).

|  | Computers | Photo | CS | Physics |
|---|---|---|---|---|
| $\mathbb{H}^{32}$ | **89.27**±**0.19** | **93.11**±**0.11** | 92.65±0.04 | **95.93**±**0.07** |
| $\mathbb{S}^{32}$ | 87.84±0.77 | 92.03±0.79 | 92.72±0.06 | 95.85±0.02 |
| $\mathbb{E}^{32}$ | 88.94±0.24 | 92.93±0.21 | **92.82**±**0.04** | 95.81±0.04 |
| $\mathbb{H}^{16} \times \mathbb{H}^{16}$ | 89.18±0.25 | 92.06±0.14 | 92.67±0.10 | 95.90±0.04 |
| $\mathbb{H}^{16} \times \mathbb{S}^{16}$ | 88.00±1.05 | 91.97±0.08 | 92.33±0.21 | 95.73±0.11 |
| $\mathbb{S}^{16} \times \mathbb{S}^{16}$ | 82.49±1.18 | 92.31±0.45 | 92.18±0.21 | 95.81±0.10 |

of representation space, and we leverage *DvM* for optimization. Manifold variants generally achieve superior results to the Euclidean one, thus verifying our motivation. On CS dataset, the performance of geometric variants is aligned with that of Euclidean and Riemannian baselines in Table 1. The proposed `MSG` is ready to switch among $\mathbb{H}$, $\mathbb{S}$, $\mathbb{E}$, and their products, matching the geometry of graphs.

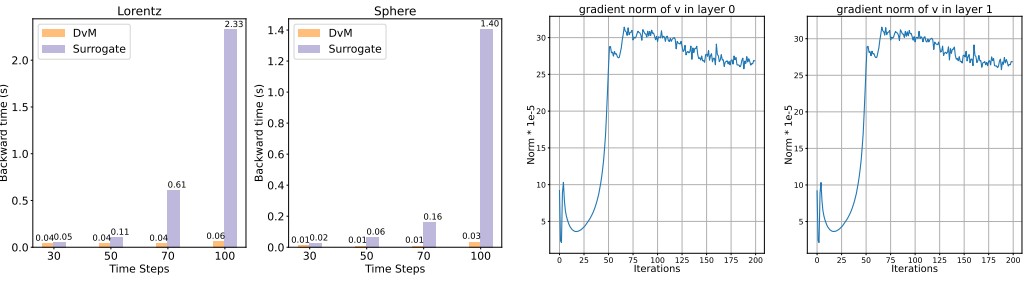

(a) Backward times in model training.    (b) Gradient norm of $L$ regarding tangent vector $\mathbf{v}$.

Figure 4: Backward time and gradient norm for node classification on Computer.

To examine the effectiveness of *DvM*, we design the optimization variant (named as Surrogate) for a given representation space. In the variant, we conduct differentiation via spikes and leverage BPTT for optimization, same as previous spiking GNNs. The training time of the optimization variants in different representation spaces are given in Fig. 4(a). Backward time of *DvM* is significantly less than that of BPTT algorithm. The reason is that *DvM* no longer needs recurrent gradient calculation of each time step (recurrence-free), while BPTT leads to high training time especially when the time step is large. In addition, we examine the backward gradient of *DvM*, and plot the gradient norm of each layer in Fig. 4(b). It demonstrates that *DvM* does not suffer from gradient vanishing/explosion.

**Energy Cost.** We investigate the energy cost of the graph models in terms of theoretical energy consumption (mJ) [5; 6], whose formula is specified in Appendix E. We summarize the results for node classification in Table 3 in which the number of parameters at the running time is listed as a reference. It shows that SNN-based models generally enjoy less energy cost than ANN-based ones.

Table 3: Energy cost. The number of parameters at the running time (KB) and theoretical energy consumption (mJ) on Computers, Photo, CS and Physics datasets. The best results are **boldfaced**, and the runner ups are underlined.

| | | Computers | | Photo | | CS | | Physics | |
|---|---|---|---|---|---|---|---|---|---|
| | | #(para.) | energy | #(para.) | energy | #(para.) | energy | #(para.) | energy |
| ANN-E | GCN [18] | 24.91 | 1.671 | 24.14 | 0.893 | 218.29 | 18.444 | 269.48 | 42.842 |
| | GAT [2] | 24.99 | 2.477 | 24.22 | 1.273 | 218.38 | 28.782 | 269.55 | 81.466 |
| | SGC [19] | **7.68** | 0.508 | **5.97** | 0.219 | **102.09** | 8.621 | **42.08** | 6.688 |
| | SAGE [1] | 49.77 | 1.671 | 48.23 | 0.893 | 436.53 | 18.444 | 538.92 | 42.842 |
| ANN-R | HGCN [3] | 24.94 | 1.614 | 24.96 | 0.869 | 217.79 | 18.390 | 269.31 | 42.800 |
| | $\kappa$-GCN [13] | 25.89 | 1.647 | 25.12 | 0.889 | 218.24 | 18.440 | 269.44 | 42.836 |
| | $\mathcal{Q}$−GCN [4] | 24.93 | 1.629 | 24.96 | 0.876 | 217.83 | 18.393 | 269.34 | 42.809 |
| | HyboNet [54] | 27.06 | 1.625 | 26.29 | 0.875 | 219.94 | 18.399 | 271.47 | 42.825 |
| SNN-E | SpikeNet [43] | 101.22 | 0.070 | 98.07 | **0.040** | 438.51 | 0.218 | 540.04 | 0.334 |
| | SpikingGCN [5] | 38.40 | 0.105 | 29.84 | 0.046 | 510.45 | 1.871 | 210.40 | 1.451 |
| | SpikeGCL [6] | 59.26 | 0.121 | 57.85 | 0.067 | 445.69 | 0.128 | 548.74 | 0.214 |
| | SpikeGT [55] | 77.07 | 1.090 | 74.46 | 0.584 | 365.28 | 6.985 | 355.77 | 12.524 |
| | MSG(Ours) | 26.95 | **0.047** | 25.68 | 0.043 | 226.15 | **0.026** | 143.72 | **0.029** |

Note, `MSG` achieves the best energy efficiency among SNN-based models except Photo dataset. In addition, it has at least $1/20$ energy cost to the Riemannian baselines.

**Visualization & Discussion.** We empirically study the connection between the proposed `MSG` and manifold ODE. In particular, we visualize a toy example of Zachary Karate Club dataset [58] on a $\mathbb{S}^1 \times \mathbb{S}^1$ in Fig. 5, where we plot each layer output on the manifold. The red curve is the path connecting the layer input and layer output, and the blue one is the direction of the geodesic. As shown in Fig. 5, the red and blue curves are coincided, that is, each layer solves an ODE describing the geodesic on the manifold.

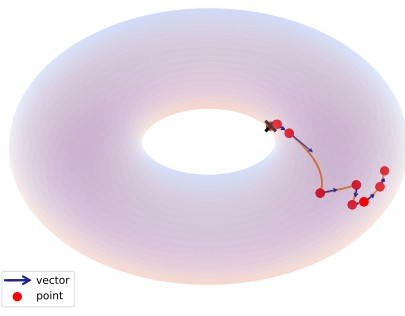

Figure 5: Visualization on $\mathbb{S}^1 \times \mathbb{S}^1$

# 7    Conclusion

In this paper, we study spiking GNN from a fundamentally different perspective of Riemannian geometry, and present a simple yet effective Manifold-valued Spiking GNN (`MSG`). Concretely, we design a manifold spiking neuron which leverages the diffeomorphism to bridge spiking representations and manifold representations. With the proposed neuron, we propose a new training algorithm with Differentiation via Manifold, which no longer needs to recur the backward gradient and thus alleviates the high latency of previous methods. An interesting theoretical result is that, `MSG` is essentially related to manifold ODE. Extensive empirical results on benchmark datasets demonstrate the superior effectiveness and energy efficiency of the proposed `MSG`.

# 8    Broader Impact and Limitations

Our work brings together two previously separate domains: spiking neural network and Riemannian geometry, and presents a novel Manifold-valued Spiking GNN for energy-efficiency graph learning, especially for the large graphs. Our work is mainly a theoretical exploration, and not tied to particular applications. A positive societal impact is the possibility of decreasing carbon emissions in training large models. None of negative societal impacts we feel must be specifically highlighted here.

**Limitation.** Our work as well as the previous spiking GNNs considers the undirected, homophilous graphs, while the spiking GNN on directed or heterophilous graphs still remains open. Also, readers may find it challenging to implement the proposed method. However, we provide downloadable code and will offer an easy-to-use interface.

## Acknowledgement

This work is supported in part by NSFC under grants 62202164 and 62322202. Philip S. Yu is supported in part by NSF under grants III-2106758, and POSE-2346158.

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

## A  Notations

We summarize the important notations of our paper in Table 4.

Table 4: Notations.

| Notation | Description |
|---|---|
| $\mathcal{M}, \mathcal{N}$ | Smooth manifolds |
| $\mathbf{x}, \mathbf{y}, \mathbf{z}$ | Points on manifolds |
| $\mathbf{o}$ | The original point on manifold |
| $T_{\mathbf{z}}\mathcal{M}$ | The tangent space at point $\mathbf{z}$ |
| $T\mathcal{M}$ | The tangent bundle of the manifold $\mathcal{M}$ |
| $u$ | The vector field over the manifold, described by an ODE |
| $T_{\mathbf{z}}^{*}\mathcal{N}$ | The dual space of $T_{\mathbf{z}}\mathcal{N}$ |
| $D_{\mathbf{z}}f$ | The differential of $f$ at point $\mathbf{z}$ |
| $(df)_{\mathbf{z}}$ | The differential $1-$form of $f$ at point $\mathbf{z}$ |
| $\{(\partial/\partial z^1)|_{\mathbf{z}}, ..., (\partial/\partial z^n)|_{\mathbf{z}}\}$ | A basis of the tangent space $T_{\mathbf{z}}\mathcal{N}$ |
| $\{(dz^1)_{\mathbf{z}}, ..., (dz^n)_{\mathbf{z}}\}$ | A basis of $T_{\mathbf{z}}^{*}\mathcal{N}$ |
| $\nabla\mathcal{L}$ | The gradient of a smooth scalar function $\mathcal{L}$ |
| $\mathrm{Exp}_{\mathbf{z}}(\cdot)$ | The exponential map at point $\mathbf{z}$ |
| $\mathrm{Log}_{\mathbf{z}}(\cdot)$ | The logarithmic map at point $\mathbf{z}$ |
| $\mathbb{S}^d$ | $d$-dimensional Sphere model of hyperspherical space |
| $\mathbb{E}^d$ | $d$-dimensional Euclidean space |
| $\mathbb{H}^d$ | $d$-dimensional Lorentz model of Hyperbolic space |
| $\mathcal{V}, \mathcal{E}, \mathbf{F}, \mathbf{A}$ | Node set $\mathcal{V}$, edge set $\mathcal{E}$, feature matrix $\mathbf{F}$ and adjacency matrix $\mathbf{A}$ |
| $\mathcal{G} = (\mathcal{V}, \mathcal{E}, \mathbf{F}, \mathbf{A})$ | A graph defined on $\mathcal{V}, \mathcal{E}, \mathbf{F}$ and $\mathbf{A}$ |
| $\Omega_i$ | Neighbourhood of node $i$ |
| $\mathcal{F}_\theta$ | A graph encoder with parameters $\theta$ |
| $V[t]$ | Membrane potential of a spiking neuron at time step $t$ |
| $H(\cdot)$ | The Heaviside step function |
| $S[t]$ | Spikes fired by a spiking neuron at time step $t$ |
| $V_{th}$ | Threshold membrane potential of a spiking neuron |
| $l$ | The index of spiking layer in neural network |
| MSNeuron | The proposed manifold spiking neuron |

## B  Proofs

In this section, we demonstrate the proofs of Theorem 4.1 (Backward Gradient) and Theorem 5.2 (MSG as Dynamic Chart Solver).

### B.1  Proof of Proposition 4.1

First, we give the formal definition of the pullback. Given a smooth map $\phi : \mathcal{M} \to \mathcal{N}$ connecting two manifolds $\mathcal{M}$ and $\mathcal{N}$, and a real, smooth function $f : \mathcal{N} \to \mathbb{R}$, the pullback of $f$ by $\phi$ is the smooth function $\phi^*f$ on $\mathcal{M}$ defined by $(\phi^*f)(\mathbf{x}) = f(\phi(\mathbf{x}))$.

Next, we introduce some properties of the pullback in the smooth manifold [59] (i.e., differential 1-form of a smooth function, communication, and pullback of a sum and a product), supporting the derivation of the backward gradient (Theorem 4.1).

**Lemma B.1** (Differential 1-form of a smooth function). *For a point $\mathbf{z} \in \mathcal{N}$ related with a coordinate chart $(U, z^1, ..., z^n)$, there is a series of covectors $\{(dz^1)_\mathbf{z}, ..., (dz^n)_\mathbf{z}\}$ forming a basis of $T_\mathbf{z}^* \mathcal{N}$ dual to the basis $\{(\partial/\partial z^1)|_\mathbf{z}, ..., (\partial/\partial z^n)|_\mathbf{z}\}$ of tangent space $T_\mathbf{z}\mathcal{N}$. Then, for any smooth function $f$ on $\mathcal{N}$ restrict to $U$, the differential 1-form of $f$ is*

$$df = \sum_{i=1}^n \frac{\partial f}{\partial z^i} dz^i. \tag{18}$$

**Lemma B.2** (Communication). *Let $F : \mathcal{N} \to \mathcal{M}$ be a smooth map, for any smooth function $g$ on $\mathcal{M}$, we have $F^*(dg) = d(F^*g)$.*

**Lemma B.3** (Pullback of a sum and a product). *Let $F : \mathcal{N} \to \mathcal{M}$ be a smooth map, $g$ is a smooth scalar function on $\mathcal{M}$, and $\omega, \gamma$ are differential 1-forms on $\mathcal{M}$. Then, we have*

$$F^*(\omega + \gamma) = F^*\omega + F^*\gamma \tag{19}$$
$$F^*(g\omega) = (F^*g)(F^*\omega). \tag{20}$$

Given the properties of the pullback, we derive the closed-form backward gradient of the real function on the manifold, and prove Theorem 4.1.

**Theorem 4.1** (Backward Gradient) *Let $\mathcal{L}$ be the scalar-valued function, and $\mathbf{z}^l$ is the output of $l$-th layer with parameter $\mathbf{W}^l$, which is delivered by tangent vector $\mathbf{v}^l$. Then, the gradient of function $\mathcal{L}$ w.r.t. $\mathbf{W}^l$ is given as follows:*

$$\nabla_{\mathbf{W}^l}\mathcal{L} = [\frac{\partial \mathbf{v}^{l-1}}{\partial \mathbf{W}^l}]^* [D_{\mathbf{v}^{l-1}}\phi^{l-1}]^* \nabla_{\mathbf{z}^l}\mathcal{L}, \quad \nabla_{\mathbf{z}^l}\mathcal{L} = [D_{\mathbf{z}^l}\psi^l]^* \nabla_{\mathbf{z}^{l+1}}\mathcal{L}, \tag{21}$$

*where $\phi^{l-1}(\cdot) = \mathrm{Exp}_{\mathbf{z}^{l-1}}(\cdot)$, $\psi^l(\cdot) = \mathrm{Exp}_{(\cdot)}(\mathbf{v}^l)$, and $[\cdot]^*$ means the matrix form of pullback.*

*Proof.* Given $\mathbf{z}^l, \mathbf{z}^{l+1}$ in $\mathcal{M}$, and $F : \mathcal{M} \to \mathcal{M}$ be the smooth map such that $F(\mathbf{z}^l) = \mathbf{z}^{l+1}$. Consider scalar loss function $L : \mathcal{M} \to \mathbb{R}$, if we relate $\mathbf{z}^l$ with a chart $(U, x^1, ..., x^m)$ and $\mathbf{z}^{l+1}$ with $(V, y^1, ..., y^m)$, the gradients of $L$ at $\mathbf{z}^{l+1}$ and $\mathbf{z}^l$ are given by Lemma. B.1,

$$\nabla_{\mathbf{z}^{l+1}}\mathcal{L} = \sum_i \frac{\partial \mathcal{L}}{\partial y^i}\bigg|_{\mathbf{z}^{l+1}} dy^i \tag{22}$$

$$\nabla_{\mathbf{z}^l}(\mathcal{L} \circ F) = \sum_i \frac{\partial \mathcal{L} \circ F}{\partial x^i}\bigg|_{\mathbf{z}^l} dx^i. \tag{23}$$

Then, we apply the pullback $F^*$ on $\nabla_{\mathbf{z}^{l+1}} L$ that

$$F^*(\nabla_{\mathbf{z}^{l+1}}\mathcal{L}) = F^* \sum_i \frac{\partial \mathcal{L}}{\partial y^i}\bigg|_{\mathbf{z}^{l+1}} dy^i|_{\mathbf{z}^{l+1}} \tag{24}$$

$$= \sum_i (F^* \frac{\partial \mathcal{L}}{\partial y^i}\bigg|_{\mathbf{z}^{l+1}})(F^* dy^i|_{\mathbf{z}^{l+1}}) \quad \text{from } \textbf{Lemma. B.3} \tag{25}$$

$$= \sum_i (\frac{\partial \mathcal{L}}{\partial y^i} \circ F)|_{\mathbf{z}^l}(d(F^* y^i)|_{\mathbf{z}^l}) \quad \text{from } \textbf{Lemma. B.2} \tag{26}$$

$$= \sum_i (\frac{\partial \mathcal{L}}{\partial y^i} \circ F)|_{\mathbf{z}^l}(d(y^i \circ F))|_{\mathbf{z}^l} \tag{27}$$

$$= \sum_i (\frac{\partial \mathcal{L}}{\partial y^i}\bigg|_{\mathbf{z}^{l+1}})(\sum_j \frac{\partial F^i}{\partial x^j} dx^j)|_{\mathbf{z}^l} \quad \text{from } \textbf{Lemma. B.1} \tag{28}$$

$$= \sum_{i,j} \frac{\partial \mathcal{L}}{\partial y^i}\bigg|_{\mathbf{z}^{l+1}} \frac{\partial F^i}{\partial x^j}\bigg|_{\mathbf{z}^l} dx^j|_{\mathbf{z}^l}, \tag{29}$$

Then, we can find that the matrix form of the pullback $F^*$ can be written as the transpose of the Jacobian matrix of $F$, denoted as $[\frac{\partial F^i}{\partial x^j}|_{\mathbf{z}^l}]^*$ or $[D_{\mathbf{z}^l}F]^*$. The derivation of $\nabla_{\mathbf{W}^l}\mathcal{L}$ is similar, we only need to use an addition process like above on $\nabla_{\mathbf{v}^{l-1}}\mathcal{L}$. $\qquad\square$

Note that, we give the closed-form expression of exponential map, logarithmic map, and parallel transport for hyperbolic and hyperspherical space, and derive the corresponding Jacobian in Sec. C.

### B.2  Proof of Theorem 5.2

**Theorem 5.2** (MSG as Dynamic Chart Solver) *If* $\mathbf{y}(t) : [\tau, \tau + \epsilon] \to \mathbb{R}^n$ *is the solution of*

$$\frac{d\mathbf{y}(t)}{dt} = (D_{\mathrm{Exp}_{\mathbf{z}}(\mathbf{y(t)})} \mathrm{Log}_{\mathbf{z}})u(\mathrm{Exp}_{\mathbf{z}}(\mathbf{y}(t)), t), \tag{30}$$

*then* $\mathbf{z}(t) = \mathrm{Exp}_{\mathbf{z}}(\mathbf{y}(t))$ *is a valid solution to the manifold ODE of Eq. (13) on* $t \in [\tau, \tau + \epsilon]$, *where* $\mathbf{z} = \mathbf{z}(\tau)$. *If* $\mathbf{y}(t)$ *is given by the first-order approximation with the* $\epsilon$ *small enough,*

$$\mathbf{y}(\tau + \epsilon) = \epsilon \cdot (D_{\mathbf{z}} \mathrm{Log}_{\mathbf{z}})u(\mathbf{z}(\tau), \tau), \tag{31}$$

*then the update process of Eqs. (4) and (8) in* MSG *is equivalent to Dynamic Chart Solver in Eq. (15).*

*Proof.* Let $t \in [\tau, \tau + \epsilon]$, then we have

$$\frac{d\mathbf{z}(t)}{dt} = (D_{\mathbf{y(t)}} \mathrm{Exp}_{\mathbf{z}})\frac{d\mathbf{y}(t)}{dt} \tag{32}$$

$$= (D_{\mathbf{y(t)}} \mathrm{Exp}_{\mathbf{z}})(D_{\mathrm{Exp}_{\mathbf{z}}(\mathbf{y(t)})} \mathrm{Log}_{\mathbf{z}})u(\mathrm{Exp}_{\mathbf{z}}(\mathbf{y}(t)), t) \tag{33}$$

$$= (D_{\mathbf{y(t)}} \mathrm{Exp}_{\mathbf{z}})(D_{\mathbf{z}(t)} \mathrm{Log}_{\mathbf{z}})u(\mathrm{Exp}_{\mathbf{z}}(\mathbf{y}(t)), t) \tag{34}$$

$$= u(\mathbf{z}(t), t). \tag{35}$$

Consider two adjacent charts $(U_1, \mathrm{Log}_{\mathbf{z}_1})$ and $(U_2, \mathrm{Log}_{\mathbf{z}_2})$, such that $\mathbf{z}_1 \in U_1$ and $\mathbf{z}_2 \in U_1 \cap U_2$. Note that, in interval $[\tau, \tau + \epsilon]$, $\mathbf{z}(\tau) = \mathbf{z}_1$ and $\mathbf{z}(\tau + \epsilon) = \mathbf{z}_2$, we have $\mathbf{y}(\tau) = \mathrm{Exp}_{\mathbf{z}_1}(\mathbf{z}_1) = \mathbf{0}$. With the first-order approximation, $\mathbf{y}(\tau + \epsilon)$ is thus given by

$$\mathbf{y}(\tau + \epsilon) = \mathbf{y}(\tau) + \epsilon \cdot (D_{\mathrm{Exp}_{\mathbf{z}_1}(\mathbf{y}(\tau))} \mathrm{Log}_{\mathbf{z}_1})u(\mathrm{Exp}_{\mathbf{z}}(\mathbf{y}(\tau)), \tau) \tag{36}$$

$$= \epsilon \cdot (D_{\mathbf{z}_1} \mathrm{Log}_{\mathbf{z}_1})u(\mathbf{z}(\tau), \tau) \tag{37}$$

Also, Eq. (37) can be treated as a step in the Euler solver [60] for a small $\epsilon$. Finally, we have $\mathbf{z}(\tau + \epsilon) = \mathrm{Exp}_{\mathbf{z}_1}(\mathbf{y}(\tau + \epsilon)) = \mathbf{z}_2$, ending the process of dynamic chart solver. That is, MSG considers the logarithmic map to define the charts, and is equivalent to Dynamic Chart Solver (Definition 5.1), completing the proof. $\square$

## C  Deviation of Jacobian

We instantiate the proposed MSG in the Lorentz model $\mathbb{H}$ of hyperbolic space, sphere model $\mathbb{S}$ of hyperspherical space, and the products of $\mathbb{H}$'s or/and $\mathbb{S}$'s. Accordingly, we derive the Jacobian in $\mathbb{H}$ and $\mathbb{S}$, and introduce the construction in the products in D.3.

### C.1  Hyperbolic Space

**Lorentz Model**  The $d$-dimensional Lorentz model $\mathbb{H}^d$ is defined on the $(d + 1)$-dimensional manifold of $\{\mathbf{z} = [z_0, z_1, \cdots, z_d]^T \in \mathbb{R}^{d+1} | \langle \mathbf{z}, \mathbf{z} \rangle_{\mathcal{L}} = -1, z_0 > 0\}$ [5], equipped with the Minkowski inner product,

$$\langle \mathbf{u}, \mathbf{v} \rangle_{\mathcal{L}} = -u_0 v_0 + \sum_{i=1}^{d} u_i v_i. \tag{38}$$

The tangent space at point $\mathbf{z} \in \mathbb{H}^d$ is $T_{\mathbf{z}}\mathbb{H}^d = \{\mathbf{v} \in \mathbb{R}^{d+1} | \langle \mathbf{z}, \mathbf{v} \rangle_{\mathcal{L}} = 0\}$, and $\mathrm{Proj}_{\mathbf{z}}(\mathbf{u}) = \mathbf{u} + \langle \mathbf{z}, \mathbf{u} \rangle_{\mathcal{L}}\mathbf{z}$ is to project a vector $u \in \mathbb{R}^{d+1}$ into the tangent space $T_{\mathbf{z}}\mathbb{H}^d$. The Lorentz norm of tangent vector is defined as $\|\mathbf{v}\|_{\mathcal{L}} = \sqrt{\langle \mathbf{z}, \mathbf{z} \rangle_{\mathcal{L}}}$.

Theorem 4.1 requires the Jocabian of $\phi(\cdot) = \mathrm{Exp}_{\mathbf{z}}(\cdot)$ and $\psi(\cdot) = \mathrm{Exp}_{(\cdot)}(\mathbf{v})$, and Lorentz model has the closed-form exponential map given as follows,

$$\mathrm{Exp}_{\mathbf{z}}(\mathbf{v}) = \cosh(\|\mathbf{v}\|_{\mathcal{L}})\mathbf{z} + \frac{\sinh(\|\mathbf{v}\|_{\mathcal{L}})}{\|\mathbf{v}\|_{\mathcal{L}}}\mathbf{v}. \tag{39}$$

---

[5]We utilize the manifold of standard curvature for model instantiation, i.e., constant curvature of $-1$ for hyperbolic space, and $1$ for hyperspherical space. Note that, hyperbolic/hyperspherical spaces of different constant curvatures are mathematically equivalent in essence.

The inverse of exponential map (i.e, the logarithmic map) is

$$\text{Log}_{\mathbf{z}}(\mathbf{x}) = \frac{\text{arcosh}(\langle \mathbf{z}, \mathbf{x} \rangle_{\mathcal{L}})}{\sinh(\text{arcosh}(\langle \mathbf{z}, \mathbf{x} \rangle_{\mathcal{L}}))}(\mathbf{x} - \langle \mathbf{z}, \mathbf{x} \rangle_{\mathcal{L}} \mathbf{z}) \tag{40}$$

**Deviation of Jacobian**   We first calculate the Jacobian of $\psi$, and it is given as

$$D_{\mathbf{z}}\psi = \cosh(\|\mathbf{v}\|_{\mathcal{L}})\mathbf{I}. \tag{41}$$

Note that, Jacobian of $\phi$ needs the Jacobian of $\|\mathbf{v}\|_{\mathcal{L}}$, which is derived as

$$D_{\mathbf{v}}\|\mathbf{v}\|_{\mathcal{L}} = \frac{d}{d\langle \mathbf{v}, \mathbf{v} \rangle_{\mathcal{L}}}(\sqrt{\langle \mathbf{v}, \mathbf{v} \rangle_{\mathcal{L}}})D_{\mathbf{v}}(\langle \mathbf{v}, \mathbf{v} \rangle_{\mathcal{L}}) = \frac{1}{\|\mathbf{v}\|_{\mathcal{L}}}\hat{\mathbf{v}}^T, \tag{42}$$

where $\hat{\mathbf{v}} = [-v_0, v_1, ..., v_d]^T$. Then, we compute the derivative of the first term of Eq. 39.

$$D_{\mathbf{v}}\cosh(\|\mathbf{v}\|_{\mathcal{L}})\mathbf{z} = \frac{d}{d\|\mathbf{v}\|_{\mathcal{L}}}(\cosh(\|\mathbf{v}\|_{\mathcal{L}}))\mathbf{z}(D_{\mathbf{v}}\|\mathbf{v}\|_{\mathcal{L}}) = \frac{\sinh(\|\mathbf{v}\|_{\mathcal{L}})}{\|\mathbf{v}\|_{\mathcal{L}}}\mathbf{z}\hat{\mathbf{v}}^T, \tag{43}$$

and the derivative of the second term is derived as

$$D_{\mathbf{v}}\frac{\sinh(\|\mathbf{v}\|_{\mathcal{L}})}{\|\mathbf{v}\|_{\mathcal{L}}}\mathbf{v} = D_{\mathbf{v}}(\frac{\sinh(\|\mathbf{v}\|_{\mathcal{L}})}{\|\mathbf{v}\|_{\mathcal{L}}})\mathbf{v} + \frac{\sinh(\|\mathbf{v}\|_{\mathcal{L}})}{\|\mathbf{v}\|_{\mathcal{L}}}D_{\mathbf{v}}\mathbf{v} \tag{44}$$

$$= \frac{\|\mathbf{v}\|_{\mathcal{L}}\cosh(\|\mathbf{v}\|_{\mathcal{L}}) - \sinh(\|\mathbf{v}\|_{\mathcal{L}})}{\|\mathbf{v}\|_{\mathcal{L}}^3}\mathbf{v}\hat{\mathbf{v}}^T + \frac{\sinh(\|\mathbf{v}\|_{\mathcal{L}})}{\|\mathbf{v}\|_{\mathcal{L}}}\mathbf{I} \tag{45}$$

Summing up above equations, we finally have

$$D_{\mathbf{v}}\phi = \frac{\|\mathbf{v}\|_{\mathcal{L}}\cosh(\|\mathbf{v}\|_{\mathcal{L}}) - \sinh(\|\mathbf{v}\|_{\mathcal{L}})}{\|\mathbf{v}\|_{\mathcal{L}}^3}\mathbf{v}\hat{\mathbf{v}}^T + \frac{\sinh(\|\mathbf{v}\|_{\mathcal{L}})}{\|\mathbf{v}\|_{\mathcal{L}}}(\mathbf{I} + \mathbf{z}\hat{\mathbf{v}}^T), \tag{46}$$

where $\mathbf{I}$ is the identity matrix.

### C.2   Hyperspherical Space

**Sphere Model**   The sphere model $\mathbb{S}^d$ is defined on the $(d+1)$-dimensional manifold of $\{\mathbf{z} = [z_0, z_1, \cdots, z_d]^T \in \mathbb{R}^{d+1} | \langle \mathbf{z}, \mathbf{z} \rangle = 1, z_0 > 0\}$ with the standard inner product $\langle \mathbf{x}, \mathbf{y} \rangle = \sum_{i=0}^{d} x_i y_i$ and norm $\|\mathbf{x}\| = \sqrt{\sum_{i=0}^{d} x_i^2}$. The tangent space at point $\mathbf{z}$ is $T_{\mathbf{z}}\mathbb{S}^d = \{\mathbf{v} \in \mathbb{R}^{d+1} | \langle \mathbf{z}, \mathbf{v} \rangle = 0\}$. Similar to Lorentz model, we have $\text{Proj}_{\mathbf{z}}(\mathbf{u}) = \mathbf{u} - \langle \mathbf{z}, \mathbf{u} \rangle \mathbf{z}$ projecting a vector $u \in \mathbb{R}^{d+1}$ into $T_{\mathbf{z}}\mathbb{S}^d$. The exponential map in the sphere model is given as

$$\text{Exp}_{\mathbf{z}}(\mathbf{v}) = \cos(\|\mathbf{v}\|)\mathbf{z} + \frac{\sin(\|\mathbf{v}\|)}{\|\mathbf{v}\|}\mathbf{v}, \tag{47}$$

and the logarithmic map is

$$\text{Log}_{\mathbf{z}}(\mathbf{x}) = \frac{\arccos(\langle \mathbf{z}, \mathbf{x} \rangle)}{\sin(\arccos(\langle \mathbf{z}, \mathbf{x} \rangle))}(\mathbf{x} - \langle \mathbf{z}, \mathbf{x} \rangle \mathbf{z}) \tag{48}$$

**Derivation of Jacobian**   We first calculate the Jacobian of $\psi$, and it is given as

$$D_{\mathbf{z}}\psi = \cos(\|\mathbf{v}\|)\mathbf{I}. \tag{49}$$

Similar to that in Lorentz model, the Jacobian of $\phi$ needs the Jacobian of $\|\mathbf{v}\|$,

$$D_{\mathbf{v}}\|\mathbf{v}\| = \frac{d}{d\langle \mathbf{v}, \mathbf{v} \rangle}(\sqrt{\langle \mathbf{v}, \mathbf{v} \rangle})D_{\mathbf{v}}(\langle \mathbf{v}, \mathbf{v} \rangle) = \frac{1}{\|\mathbf{v}\|}\mathbf{v}^T. \tag{50}$$

Then, we compute the derivative of the first term of Eq. 47.

$$D_{\mathbf{v}}\cos(\|\mathbf{v}\|)\mathbf{z} = \frac{d}{d\|\mathbf{v}\|}(\cos(\|\mathbf{v}\|))\mathbf{z}(D_{\mathbf{v}}\|\mathbf{v}\|) = \frac{-\sin(\|\mathbf{v}\|)}{\|\mathbf{v}\|}\mathbf{z}\mathbf{v}^T, \tag{51}$$

and the derivative of the second term is

$$D_{\mathbf{v}}\frac{\sin(\|\mathbf{v}\|)}{\|\mathbf{v}\|}\mathbf{v} = D_{\mathbf{v}}(\frac{\sin(\|\mathbf{v}\|)}{\|\mathbf{v}\|})\mathbf{v} + \frac{\sin(\|\mathbf{v}\|)}{\|\mathbf{v}\|}D_{\mathbf{v}}\mathbf{v} \tag{52}$$

$$= \frac{\|\mathbf{v}\|\cos(\|\mathbf{v}\|) - \sinh(\|\mathbf{v}\|)}{\|\mathbf{v}\|^3}\mathbf{v}\mathbf{v}^T + \frac{\sin(\|\mathbf{v}\|)}{\|\mathbf{v}\|}\mathbf{I} \tag{53}$$

Summing up above equations, we finally have

$$D_{\mathbf{v}}\phi = \frac{\|\mathbf{v}\|\cos(\|\mathbf{v}\|_{\mathcal{L}}) - \sin(\|\mathbf{v}\|)}{\|\mathbf{v}\|^3}\mathbf{v}\mathbf{v}^T + \frac{\sin(\|\mathbf{v}\|)}{\|\mathbf{v}\|}(\mathbf{I} - \mathbf{z}\mathbf{v}^T). \tag{54}$$

# D  Riemannian Geometry

## D.1  Some Notations

Here, we give the formal descriptions of the notions mentioned in the main paper, and please refer to [61] for systematic elaborations.

**Geodesically Complete Manifold.**  A manifold is said to be geodesically complete if the maximal defining interval of any geodesic is $\mathbb{R}$. For any Riemannian manifold $(\mathcal{M}, g)$ admitting a metric structure given by the length of geodesic

$$d(p, q) = \inf\{L(\gamma)|\gamma \text{ is a piecewise smooth curve connecting } p \text{ to } q\}, \qquad (55)$$

the completeness of $d$ can be described as a metric space is complete if any Cauchy sequence in it converges. For instance, hyperbolic space as well as hyperspherical space is geodesically complete.

**Tangent Bundle.**  Given an $n$-dimensional smooth manifold $\mathcal{M}$, the tangent bundle $T\mathcal{M}$ is the disjoint union of all the tangent spaces of the manifold $T\mathcal{M} = \bigsqcup_{\mathbf{z} \in \mathcal{M}} T_{\mathbf{z}}\mathcal{M}$, and the tangent bundle with the projection $\pi(\mathbf{v}) = \mathbf{p}$ for all $\mathbf{v} \in T_{\mathbf{p}}\mathcal{M}$ is a vector bundle of rank $n$.

**Chart.**  A chart of a manifold is a pair $(U, \phi)$ where $U$ is an open set in the manifold and $\phi : U \to \mathbb{R}^n$ is homeomorphism onto it image, giving a local coordinate of the manifold. In other words, it provides a way of identifying the manifold locally with a Euclidean space. Given two charts $(U_1, \phi_1)$ and $(U_2, \phi_2)$, if the overlap

$$\phi_2 \circ \phi_1^{-1} : \phi_1(U_1 \cap U_2) \to \phi_2(U_1 \cap U_2) \text{ and } \phi_1 \circ \phi_2^{-1} : \phi_2(U_1 \cap U_2) \to \phi_1(U_1 \cap U_2), \qquad (56)$$

the two charts are said to be compatible.

**Curvature and Sectional Curvature.**  The curvature is a notion describing the extent of how a manifold derivatives from being "flat". In particular, the curvature of a Riemannian manifold $\mathcal{M}$ should be viewed as a measure $R(X, Y)Z$ of the extent to which the operator $(X, Y) \to \nabla_X \nabla_Y Z$ is symmetric, where $\nabla$ is a connection on $\mathcal{M}$ (where $X, Y, Z$ are vector fields, with $Z$ fixed). Sectional curvature is simpler object of curvature and is defined on two independent vector unit in the tangent space. When $\nabla$ is the Levi-Civita connection induced by a Riemannian metric on $\mathcal{M}$, it turns out that the curvature operator $R$ can be recovered from the sectional curvature.

**Constant Curvature Space, Hyperbolic Space, Hyperspherical Space.**  A Riemannian manifold is said to be a constant curvature space (CCS) if the sectional curvature is constant scalar everywhere on the manifold. When the CCS has a negative constant curvature, it is referred to as hyperbolic space, and the CCS is hyperspherical when its constant curvature is positive.

## D.2  $\kappa$-stereographic model and Stereographic Projection

$\kappa$-**stereographic model**  It gives a unified formalism for both positive and negative constant curvatures. For a positive curvature, it is the hyperspherical model for the hyperspherical space, and for a negative curvature, it switches to the Poincaré ball model.

Specifically, for a curvature $\kappa$ and a dimension $d \geq 2$, the $\kappa$-stereographic model $\mathfrak{st}_\kappa^d$ is defined on the manifold of $\{\mathbf{x} \in \mathbb{R}^d | -\kappa\|\mathbf{x}\|^2 < 1\}$, which is equipped with a Riemannian metric $\mathfrak{g}_{\mathbf{x}}^\kappa = \frac{4}{(1+\kappa\|\mathbf{x}\|^2)^2}\mathbf{I}$ for any constant curvature $\kappa$. When $\kappa \geq 0$, the defining domain is $\mathbb{R}^d$ in which the stereographic projection of the Sphere model of hyperspherical space is endowed. When $\kappa < 0$, the manifold $\mathfrak{st}_\kappa^d$ is represented in an open ball of radius $\frac{1}{\sqrt{-\kappa}}$, and is the stereographic projection of the Lorentz model of hyperbolic space.

The $\kappa$-stereographical model is a gyrovector space in which a non-associative vector operator system is defined. For $\mathbf{x}, \mathbf{y} \in \mathbb{G}^n$, $a \in \mathbb{R}$, the $\kappa$-addition (a.k.a. Möbius addition) is given as

$$\mathbf{x} \oplus_\kappa \mathbf{y} = \frac{(1 - 2\kappa\mathbf{x}^T\mathbf{y} - \kappa\|\mathbf{y}\|^2)\mathbf{x} + (1 + \kappa\|\mathbf{x}\|^2)\mathbf{y}}{1 - 2\kappa\mathbf{x}^T\mathbf{y} + \kappa^2\|\mathbf{x}\|^2\|\mathbf{y}\|^2} \qquad (57)$$

The distance function given by $\kappa$-addition is thus formulated as

$$d_\kappa(\mathbf{x}, \mathbf{y}) = 2 \tan_\kappa^{-1}(\|(-\mathbf{x}) \oplus_\kappa \mathbf{y}\|) \tag{58}$$

The $\kappa$-scaling for any real scalar $c$ is defined as

$$c \otimes_\kappa \mathbf{x} = \tan_\kappa(c \cdot \tan_\kappa^{-1}(\|\mathbf{x}\|)) \frac{\mathbf{x}}{\|\mathbf{x}\|} \tag{59}$$

The unit-speed geodesic from $\mathbf{x}$ to $\mathbf{y}$ is

$$\gamma_{\mathbf{x} \to \mathbf{y}}(t) = \mathbf{x} \oplus_\kappa (t \otimes_\kappa ((-\mathbf{x}) \oplus_\kappa \mathbf{y})) \tag{60}$$

With the unit-speed geodesic, the exponential map as well as its inverse (i.e., the logarithmic map) has the closed-form expression as follows,

$$\mathrm{Exp}_\mathbf{x}^\kappa(\mathbf{v}) = \mathbf{x} \oplus_\kappa \left( \tan_\kappa(|\kappa|^{\frac{1}{2}} \frac{\lambda_\mathbf{x}^\kappa \|\mathbf{v}\|}{2}) \frac{\mathbf{v}}{\|\mathbf{v}\|} \right) \tag{61}$$

$$\mathrm{Log}_\mathbf{x}^\kappa(\mathbf{y}) = \frac{2|\kappa|^{-\frac{1}{2}}}{\lambda_\mathbf{x}^\kappa} \tan_\kappa^{-1}(\|(-\mathbf{x}) \oplus_\kappa \mathbf{y}\|) \frac{(-\mathbf{x}) \oplus_\kappa \mathbf{y}}{\|(-\mathbf{x}) \oplus_\kappa \mathbf{y}\|}, \tag{62}$$

where the curvature-aware trigonometric function is utilized, e.g.,

$$\tan_\kappa(x) = \begin{cases} \frac{1}{\sqrt{\kappa}} \tan(x) & \kappa > 0, \\ x & \kappa = 0, \\ \frac{1}{\sqrt{-\kappa}} \tanh(x) & \kappa < 0. \end{cases} \tag{63}$$

**Stereographic Projection**   The stereographic projection is a diffeomorphism connecting the different model spaces of Riemannian manifold. In particular, it is defined as a map $\pi : \mathbb{L}_\kappa^d/\mathbb{S}_\kappa^d \to \mathfrak{st}_\kappa^d$ taking the form of

$$\pi : \mathbb{L}_\kappa^d/\mathbb{S}_\kappa^d \to \mathfrak{st}_\kappa^d, \quad \mathbf{x} = \frac{1}{1 + \sqrt{|\kappa|}\mathbf{x}_{d+1}'} \mathbf{x}_{1:d}', \tag{64}$$

where $\mathbf{x}'$ is a point on the Lorentz model $\mathbb{L}_\kappa^d$ or Sphere model $\mathbb{L}_\kappa^d$, and $\mathbf{x}$, the image of the projection, is the corresponding point in the gyrovector ball of $\kappa-$stereographic model. The inverse projection is given as follows,

$$\pi^{-1} : \mathfrak{st}_\kappa^d \to \mathbb{L}_\kappa^d/\mathbb{S}_\kappa^d, \quad \mathbf{x}' = \left( \lambda_\mathbf{x}^\kappa \mathbf{x}, \frac{1}{\sqrt{|\kappa|}}(\lambda_\mathbf{x}^\kappa - 1) \right), \tag{65}$$

where $\lambda_\mathbf{x}^\kappa = \frac{2}{1+\kappa\|\mathbf{x}\|^2}$ is known as the conformal factor.

### D.3   Cartesian Product and Product Manifold

The concept of product manifolds allows for creating a new manifold from a finite collection of existing ones. Given a set of smooth manifolds $\mathcal{M}_1, \mathcal{M}_2, \ldots, \mathcal{M}_k$, the product manifold $\mathbb{P}$ is given as the Cartesian product of these manifolds:

$$\mathbb{P} = \mathcal{M}_1 \times \mathcal{M}_2 \times \ldots \times \mathcal{M}_k, \tag{66}$$

where $\otimes$ denotes the Cartesian product. Specifically, with the Cartesian product construction, a point $\mathbf{x} \in \mathbb{P}$ are represented by a concatenation of $\mathbf{x} = [\mathbf{x}_1, \ldots, \mathbf{x}_k]$, where $\mathbf{x}_i \in \mathcal{M}_i$. A tangent vector $\mathbf{v} \in T_\mathbf{x}\mathbb{P}$ at a point $\mathbf{x}$ can be given as $\mathbf{v} = [\mathbf{v}_1, \ldots, \mathbf{v}_k]$, where $\mathbf{v}_i \in T_{\mathbf{x}_i}\mathcal{M}_i$. If each manifold $\mathcal{M}_i$ is equipped with a metric tensor $\mathbf{g}_i$, the product metric $\mathbf{g}$ decomposes into the direct sum of the individual metrics $\mathbf{g} = \oplus_{i=1}^k \mathbf{g}^i$, which can be expressed as $\mathrm{Diag}(\mathbf{g}^1, ..., \mathbf{g}^k)$. For $\mathbf{x}$ and $\mathbf{y} \in \mathbf{P}$, the distance between them is defined as $d_\mathbb{P}(\mathbf{x}, \mathbf{y}) = \sum_{i=1}^k d_{\mathcal{M}_i}(\mathbf{x}_i, \mathbf{y}_i)$. Accordingly, the exponential map is given as

$$\mathrm{Exp}_\mathbf{x}([\mathbf{v}_1, \ldots, \mathbf{v}_k]) = \left[ \mathrm{Exp}_{\mathbf{x}_1}(\mathbf{v}_1), \mathrm{Exp}_{\mathbf{x}_2}(\mathbf{v}_2), \ldots, \mathrm{Exp}_{\mathbf{x}_k}(\mathbf{v}_k) \right]. \tag{67}$$

Table 5: Dataset statitics.

|  | Computers | Photo | CS | Physics |
|---|---|---|---|---|
| #Nodes | 13752 | 7650 | 18,333 | 34,493 |
| #Features | 767 | 745 | 6,805 | 8,415 |
| #Edges | 245861 | 119081 | 163,788 | 495,924 |
| #Classes | 10 | 8 | 15 | 5 |

# E    Experimental Setups

## E.1    Dataset

We use four common benchmark datasets to evaluate our model and Table 5 show the details of the datasets. There are two co-purchase graphs including Amazon-Photo and Amazon-Computers [53] and two co-author network including CS and Physics [53].

## E.2    Baselines

Table 6: The categories of the baselines

|  | ANN-based Models | SNN-based Models |
|---|---|---|
| **Euclidean Space** | GCN [18], SAGE [1], GAT [2], SGC [19] | SpikeNet [45], SpikeGraphormer [55], SpikeGCN [5], SpikeGCL [6] |
| **Riemannian Space** | $Q-$GCN [4], $\kappa-$GCN [13], HyboNet [54], HGCN [3] |  |

As shown in Table 6, we divide the baselines into three categories: ANN-based Euclidean GNNs, ANN-based Riemannian GNNs and SNN-based Euclidean GNNs. Note that, none of the existing work studies the SNN-based GNN in Riemannian space, to the best of our knowledge.

**ANN-based Euclidean GNNs**

- **GCN** [18]: It defines graph convolution on the spectral domain.
- **SAGE** [1]: It gives the aggregate-and-combine formulation for the message passing over the graph.
- **GAT** [2]: It introduces the attention mechanism for the learning on graphs.
- **SGC** [19]: It reformulates GCN [18] with feature propagation and linear layer, acting as a low-pass filter.

**ANN-based Riemannian GNNs**

- **HGCN** [3] : It generalizes GAT [2] in the Lorentz model of hyperbolic space in which the graph convolution is conducted in the tangent space.
- $\kappa-$**GCN** [13]: It generalizes GCN [18] to the $\kappa-$stereographical model of constant curvature spaces where several gyrovector operators are given in the unified formalism.
- **HyboNet** [54]: It introduces a parameterized Lorentz transformation for hyperbolic graph modeling without the tangent space.
- $Q-$**GCN** [4]: It studies the graph convolution network in the Pseudo-Riemannian manifold.

**SNN-based Euclidean GNNs.**

- **SpikeGCN** [5]: It integrates SNN and graph convolution network in which SNN acts as an activation function.
- **SpikeGraphormer** [55] (termed as SpikeGT for short in our main paper): It generalizes a kind of graph transformer with spiking neurons, accompanied by an ANN for improving the performance.

- **SpikeGCL** [6]: It introduces a method to perform graph contrastive learning with the spiking GNN.
- **SpikeNet** [45]: It is original designed for dynamic graph modeling, and we utilize its version for static graph following [6].

Note that, existing SNN-based GNNs work with Euclidean space, and leverage the BPTT training with surrogate gradient, suffering from high latency.

### E.3 Theoretical Energy Consumption

Following the previous works [6; 5; 62], we calculate the theoretical energy consumption for each model, instead of measuring actual electricity usage, for fair comparison.

- For the SNN-based models, the energy consumption involves encoding energy $E_{\text{encoding}}$ and spiking process energy $E_{\text{spiking}}$. The former is calculated by the number of multiply-and-accumulate (MAC) operations, and the latter is given by the number of SOP operations. The energy consumption is thus defined as follows,

$$E = E_{\text{encoding}} + E_{\text{spiking}} = E_{\text{MAC}} \sum_{t=1}^{T} Nd S_t + E_{\text{SOP}} \sum_{t=1}^{T} \sum_{l=1}^{L} S_t^l, \tag{68}$$

where the scaling constant $E_{\text{MAC}}$ and $E_{\text{SOP}}$ are set to $4.6pJ$ and $3.7pJ$, respectively. $N$ is the number of nodes in the graph, $d$ is the dimension of node features, $L$ is the number of layers in the neural model. $T$ is the time steps of the spikes, and $S_t^l$ denotes the output spikes at time step $t$ and layer $l$.

- For the ANN-based models, the energy consumption is given by embedding generation step and aggregation step, and both of them are calculated by MAC operations. In the embedding generation step, the feature transformation with a weight matrix of $\mathbb{R}^{d_{in} \times d_{out}}$ executes $Nd_{in}d_{out}$ multiplication and $Nd_{in}d_{out}$ addition operations. In the aggregation step, $|\mathcal{E}|d_{in}$ is the number of multiplication and $|\mathcal{E}|d_{out}$ is the number of addition operations. Supposing $|\mathcal{E}|d_{in} = |\mathcal{E}|d_{out}$, the energy consumption is given as follows,

$$E = E_{\text{MAC}}(Nd_{in}d_{out} + |\mathcal{E}|d_{out}). \tag{69}$$

where the scaling constant $E_{\text{MAC}}$ is set to $4.6pJ$, $|\mathcal{E}|$ is the number of edge, $d_{in}$ and $d_{out}$ are the input and output dimensions.

### E.4 Implementation Notes

**Model Instantiation.** The proposed `MSG` is instantiated in the Lorentz model $\mathbb{H}$ of hyperbolic space or Sphere model $\mathbb{S}$ of hyperspherical space as well as the products over $\mathbb{H}$ and $\mathbb{S}$. Note that, `MSG` can be equivalently instantiated on the $\kappa-$stereographic model (i.e., Poincaré model of hyperbolic space or Hyperspherical model of hyperspherical space), given the closed form Riemannian metric and exponential map. The equivalence can be achieved by scaling the stereographical projection. We leverage the Lorentz model $\mathbb{H}$ by default.

**Hyperparameters.** The dimension of the manifold is set as 32. When we instantiate `MSG` on the product manifold, the sum of factor manifold's dimensions is defined as 32. The manifold spiking neuron is based on the IF model [49] by default, and it is ready to switch to the LIF model [49]. The time latency $T$ for neurons is set to 5 or 15. The step size $\epsilon$ in Eq. 8 is set to 0.1. The hyperparameters are tuned with grid search, in which the learning rate is $\{0.01, 0.003\}$ for node classification and $\{0.003, 0.001\}$ for link prediction, and the dropout rate is in $\{0.1, 0.3, 0.5\}$.

**Hardware.** Experiments are conducted on the NVIDIA GeForce RTX 4090 GPU 24GB memory, and AMD EPYC 9654 CPU 315 with 96-Core Processor.

## F  Additional Results

In this section, we show the additional results on backward gradient, comparison between IF and LIF model, link prediction and visualization.

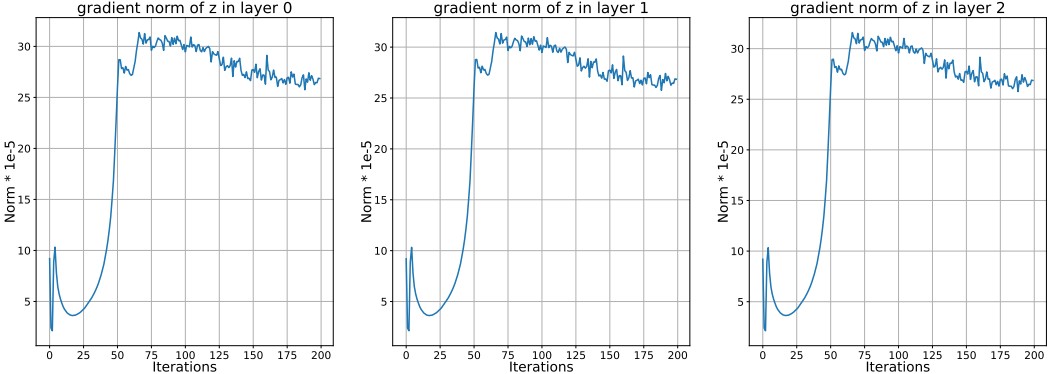

(a) The norm of backward gradient of $L$ with respect to $\mathbf{z}$ in each spiking layers.

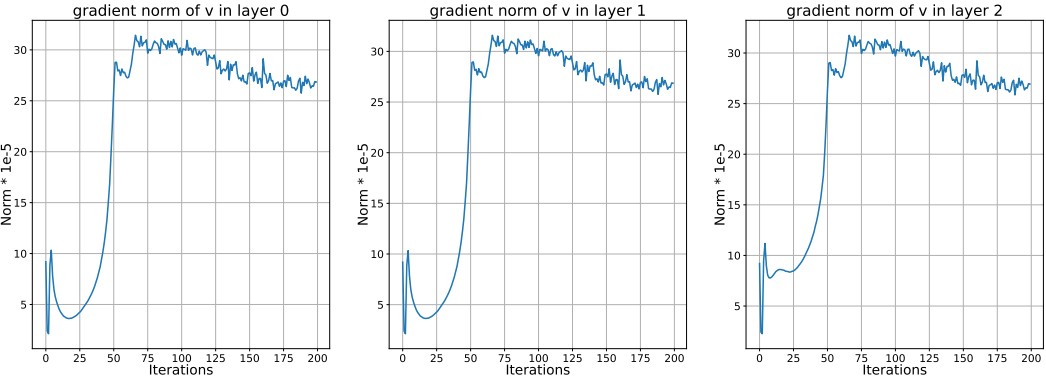

(b) The norm of backward gradient of $L$ with respect to $\mathbf{v}$ in each spiking layers.

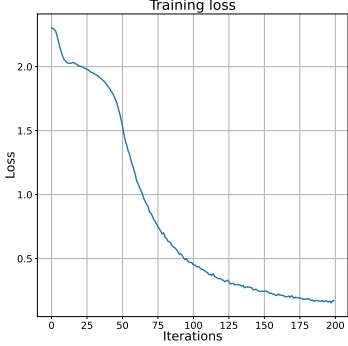

(c) The loss in model training.

Figure 6: Visualizations of the training process for node classification on Computer dataset.

**Backward Gradient.**    Previous studies compute backward gradients though the Differentiation via Spikes (*DvS*). Distinguishing from the previous studies, we compute backward gradients though the Differentiation via Manifold (*DvM*). In order to examine the backward gradients, we visualize the training process for node classification on Computer dataset. Concretely, we plot the norm of backward gradients in each iteration in Figs. 6 (a) and (b) together with the value of loss function in Figs. 6 (c). As shown in Fig. 6, the proposed algorithm with *DvM* converges well, and *the backward gradients do not suffer from gradient vanishing or gradient explosion.*

**Comparison between IF and LIF model.**    In the main paper, the proposed `MSG` is built with the IF model, and it is applicable to LIF model as well. We compare the performance between IF and LIF model in different algorithms (*DvM* and *DvS*) and in different manifolds (hyperbolic $\mathbb{H}$, hyperspherical $\mathbb{S}$, Euclidean $\mathbb{E}$ and the product spaces among $\mathbb{H}$ and $\mathbb{S}$) for a comprehensive evaluation.

The results of node classification on Computer, Photo, CS and Physics datasets are summarized in Table 7 and Table 8. Note that, *IF model and LIF model achieves competitive performance in every case. We opt for IF model in the model instantiation for simplicity.*

Table 7: Comparison between IF and LIF model in Node Classification, qualified by classification accuracy (%). The proposed model is trained by Differentiation via Spikes (i.e., BPTT with the surrogate gradient).

|  |  | Computers | Photo | CS | Physics |
|---|---|---|---|---|---|
| IF | $\mathbb{H}^{32}$ | 89.65±0.18 | 93.46±0.12 | 91.73±0.34 | 95.16±0.17 |
|  | $\mathbb{S}^{32}$ | 89.37±0.26 | 93.39±0.21 | 91.59±0.24 | 95.15±0.10 |
|  | $\mathbb{E}^{32}$ | 88.36±0.95 | 92.75±0.40 | 92.53±0.06 | 96.00±0.03 |
| LIF | $\mathbb{H}^{32}$ | 89.58±0.34 | 92.81±0.21 | 92.44±0.13 | 95.63±0.02 |
|  | $\mathbb{S}^{32}$ | 89.32±0.19 | 92.82±0.15 | 92.11±0.16 | 95.54±0.04 |
|  | $\mathbb{E}^{32}$ | 89.13±0.27 | 92.93±0.23 | 92.56±0.15 | 95.97±0.05 |

Table 8: Comparison between IF and LIF model in Node Classification, qualified by classification accuracy (%). The proposed model is trained by Differentiation via Manifold.

|  |  | Computers | Photo | CS | Physics |
|---|---|---|---|---|---|
| IF | $\mathbb{H}^{32}$ | 89.27±0.19 | 93.11±0.11 | 92.65±0.04 | 95.93±0.07 |
|  | $\mathbb{S}^{32}$ | 87.84±0.77 | 92.03±0.79 | 92.72±0.06 | 95.85±0.02 |
|  | $\mathbb{E}^{32}$ | 88.94±0.24 | 92.93±0.21 | 92.82±0.04 | 95.81±0.04 |
| LIF | $\mathbb{H}^{32}$ | 88.71±0.13 | 92.74±0.07 | 92.43±0.11 | 95.64±0.06 |
|  | $\mathbb{S}^{32}$ | 88.43±0.10 | 92.45±0.13 | 92.53±0.06 | 95.84±0.03 |
|  | $\mathbb{E}^{32}$ | 86.34±0.19 | 92.42±0.09 | 92.66±0.09 | 96.02±0.03 |

**Link Prediction.** The performance of link prediction in terms of AUC is provided in the main paper. We show the results on Computer, Photo, CS and Physics datasets in terms of AP (%) in Table 9. In particular, we feed the spiking representation of SpikingGCN and SpikeGCL into the Fermi-Dirac decoder same as the proposed MSG, while SpikeNet and SpikeGT are designed for node classification specially. As shown in Table 9, *the proposed spiking MSG consistently achieves the best results among the spiking GNNs on all the four datasets, and even achieves the competitive performances with the strong Riemannian baselines.*

Table 9: Link Prediction in terms of AP (%) on Computers, Photo, CS and Physics datasets. The best results are **boldfaced**, and the runner-ups are underlined. The standard derivations are given in the subscripts. OOM denotes Out-Of-Memory.

|  |  | Computers | Photo | CS | Physics |
|---|---|---|---|---|---|
| ANN-E | GCN [18] | 92.10±0.50 | 86.43±0.40 | 93.38±0.92 | 92.83±0.47 |
|  | GAT [2] | 91.61±0.65 | 87.04±0.05 | 94.34±0.60 | 93.44±0.45 |
|  | SGC [19] | 90.78±0.60 | 90.05±0.70 | 95.34±0.58 | 95.37±0.81 |
|  | SAGE [1] | 90.51±0.42 | 88.40±0.40 | 94.86±0.21 | 95.15±0.51 |
| ANN-R | HGCN [3] | **96.46±0.74** | 93.86±0.30 | 91.90±0.35 | 92.01±0.57 |
|  | $\kappa$-GCN [13] | 94.80±0.60 | 93.50±0.09 | 94.97±0.07 | 94.16±0.48 |
|  | $\mathcal{Q}$-GCN [4] | 96.28±0.03 | 96.65±0.10 | 92.24±0.75 | OOM |
|  | HyboNet [54] | 95.78±0.07 | **96.79±0.04** | **96.21±0.33** | **98.12±0.97** |
| SNN-E | SpikeNet [45] | - | - | - | - |
|  | SpikingGCN [5] | 91.17±1.64 | 93.16±0.04 | 94.79±1.23 | 92.19±0.90 |
|  | SpikeGCL [6] | 92.54±0.03 | 95.16±0.12 | 95.06±0.19 | 91.82±0.25 |
|  | SpikeGT [55] | - | - | - | - |
|  | MSG (Ours) | 94.45±0.78 | 96.46±0.19 | 95.12±0.12 | 92.53±0.19 |

**Visualization.** Here, we visualize the forward pass of the proposed MSG and empirically demonstrate the connection between MSG and manifold ordinary differential equation (ODE).

We choose a toy example of KarateClub dataset. The proposed MSG are instantiated on the 2D manifold for the ease of visualization. Specifically, we plot the node representation of each spiking

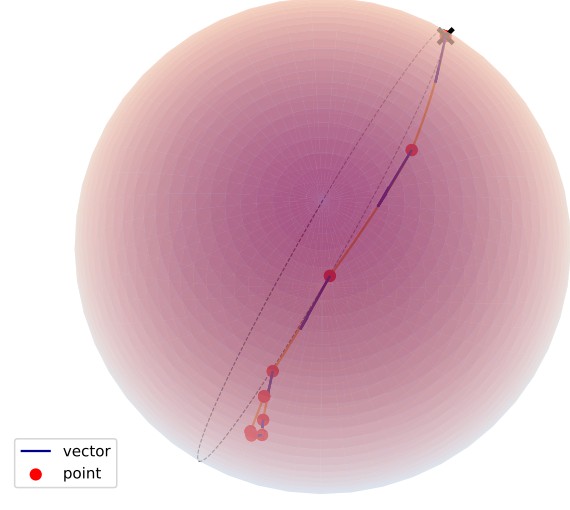

(a) 1-th nodes.

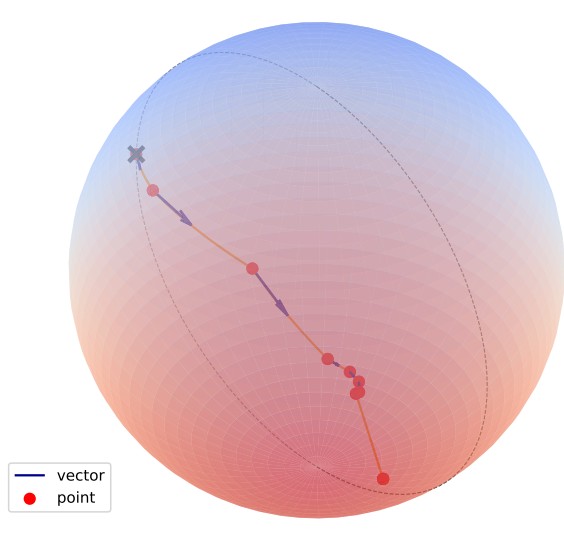

(b) 17-th nodes.

Figure 7: Visualizations of node representations on Zachary karateClub datasets [58].

layer in Fig. 7(a) and Fig. 7(b) in which the curve connecting the outputs of successive layer is marked in red, and blue arrow is the direction of the geodesic. It is shown that a spiking layer forwards the node along the geodesic on the manifold. In other words, *each layer, constructing a chart given by the exponential map, is a solver of the ODE describing the geodesic.*

