# OpenReview forum: "Spiking Graph Neural Network on Riemannian Manifolds"
_NeurIPS.cc/2024/Conference — NeurIPS 2024 poster_

### Official Review · Reviewer_hcr5 · 2024-07-01

**Soundness:** 3
**Presentation:** 3
**Contribution:** 4
**Rating:** 8
**Confidence:** 4

**Summary:**

The authors generalize spiking GNN to Riemannian manifold, and design a new architecture of parallel forwarding so as to boost model training. Then, the proposed MSG is evaluated with 12 baselines on real graphs.

**Strengths:**

S1. A technical strong paper, and No error is detected. It presents a Riemannian optimization to boost the SNN training and connects the proposed model to Riemannian ODE.

S2. The studied problem of Spiking GNN training is interesting, since it is time consuming to conduct BPTT training of SGNN especially when the spiking train is long.

S3. The authors give a new architecture of parallel forwarding so that graph is modeled in Riemannian manifold, different from the previous SGNN in Euclidean space.

S4. The experiment and visualization are convincing, and the results show that Riemannian modeling of SNN generally achieves superior performance.

**Weaknesses:**

W1. The proposed model works with the backbone of GCN, and the performance of other backbones, GAT for instance, is not examined.

W2. Some experimental details are not specified, e.g., the link prediction setting with SNN baselines, which is not frequently reported in graph SNN studies.

**Questions:**

Q1. The authors add a parallel Riemannian forwarding to accelerate the SNN training, leveraging DvM. However, it needs to specify the additional parameter and will it damage the energy efficiency of SNN?

Q2. In the ablation study of Section 6.2, the training time of DvM is much less than that of BPTT, and the question is that, is DvM’s backward time of independent from the steps of spikes?

Q3. Please specify the manifold of Fig. 5 in the Experiment, i.e., the construction of the manifold or its equation.

Q4. What is the different between this paper and the recent paper, Continuous spiking graph neural networks, regarding ODE?

**Limitations:**

The authors specify the limitations and potential negative social impact in Sec. 8.

---

> ### Author Rebuttal · Authors · 2024-08-05
>
> **W1: Performance on GAT backbone.**
>
> The proposed MSG is applicable to any backbone GNN, where the incoming current $x$ is given by the corresponding backbone.
>
> We show the results of GAT backbone in the sphere manifold on Computers, Photo, CS and Physics datasets as follows. (The mean with standard derivations of 10 independent runs are reported.)
>
> |         | Computers   | Photo       |  CS         | Physics     |
> |---------|-------------|-------------|-------------|-------------|
> | GAT   | 86.82±0.04  | 86.68±1.32  | 91.74±0.22  | 95.11±0.29  |
> | $\mathbb{S}^{32}$(MSG GCN) | 87.84±0.77  | 92.03±0.79  | 92.72±0.06  | 95.85±0.02  |
> | $\mathbb{S}^{32}$(MSG GAT)| 88.51±0.88  | 92.63±0.45  | 92.51±0.08  | 95.82±0.03  |
>
> As shown in the table above, MSG achieves competitive performance on GAT and GCN backbones, and consistently outperforms the original GAT methods.
>
> **W2: Experimental settings of SNN baselines for link prediction**
>
> For the SNNs generating spiking representations (i.e., SpikeGCN and SpikeGCL), we feed the spiking representation into Fermi-Dirac decoder, obtaining the pairwise similarity for link prediction.
>
> For the SNNs that cannot generate spiking representations (i.e., SpikeNet and SpikeGraphormer), we do not report the results of link prediction, since there exits no well-recognized similarity for the spikes.
>
> **Q1: The additional parameters and energy consumption by adding Riemannian forwarding.**
>
> The proposed MSG does not introduce additional parameters in the Riemannian forwarding.
>
> The proposed MSG leverages the tangent vector, rather than the manifold representations, for model inference. Thus, only limited float-point operations are involved.
>
> We report the theoretical energy consumption (mJ) in Table 3and, in fact, MSG presents competitive and even better energy efficiency to the previous Spiking GNNs, and obtain superior accuracy.
>
> **Q2: In model training, is DvM’s backward time of independent from the steps of spikes?**
>
> The gradient backpropagation is introduced in Theorem 4.1, and DvM relates to the step of spikes ONLY in Eq. (7), or more specifically, the derivate of $\frac{\partial \mathbf{v}_i}{\partial {\mathbf{x}_i[t]}}$.
> Note that, the pooling of Eq. (7) is the mean of $x$ sequence, and the derivative is simple and very fast to compute.
>
> In contrast, the traditional BPTT methods recursively compute the whole backpropagation process for each time step.
>
> Consequently, the proposed DvM achieves the significant less training time to BPTT, as shown in Fig 4.
>
> **Q3: Specify the manifold in Figure 5.**
>
> Fig. 5 is a torus, generated by rotating a circle around an axis coplanar with the circle.
>
> The equation of the torus is specified as follows,
> $$
> \begin{array}{lll}
> x(\theta, \phi) = (R + r \cos\theta) \cos\phi, \\\\
> y(\theta, \phi) = (R + r \cos\theta) \sin\phi, \\\\
> z(\theta, \phi) = r \sin\theta,
> \end{array}
> $$
> where $x(\theta, \phi)$, $y(\theta, \phi)$, and $z(\theta, \phi)$ represent the three-dimensional coordinates of any point on the surface of the torus.
>
> $R$ is the radius of the major circle, which is the distance from the center of the torus to the center of the minor circle.
>
> $r$ is the radius of the minor circle, which is the distance from the center of the minor circle to the surface of the torus.
>
> $\theta$ is the angle parameter for points on the minor circle, used to describe the position of points on the minor circle.
>
> $\phi$ is the angle parameter for the position of the minor circle's center on the major circle, used to describe the location of the minor circle center along the path of the major circle.
>
> Equivalently, a torus is constructed as the product of two circles: $S^1 \times S^1$.
>
> **Q4: The difference between this paper and the recent paper (Continuous spiking graph neural networks) regarding ODE.**
>
> ''Continuous spiking graph neural networks''' is the reference [11] of our paper. [11] study the connections between spiking neural network, continuous GNN, and ODE in Euclidean space, while we relate GNN and Manifold ODE, as stated in Sec. 5.
>
> More specifically, [11] establishes ODE w.r.t. to the **spiking** representations, while we consider the ODE w.r.t. the **manifold** representations, and the spikes are related to the Charts as elaborated in Sec. 5.

---

### Official Review · Reviewer_RDnD · 2024-07-09

**Soundness:** 3
**Presentation:** 3
**Contribution:** 3
**Rating:** 6
**Confidence:** 4

**Summary:**

This work proposes a spiking graph neural network on Riemannian manifolds, named Manifold-valued Spiking GNN (MSG). This work also develops a new training algorithm of differentiation via manifold (DvM) that avoids the high training overhead of BPTT methods and proves that the MSG is a neural ODE solver. Experimental results show that MSG outperforms existing spiking GNNs with low energy consumption.

**Strengths:**

1. It is a novel idea to explore spiking GNN on Riemannian manifolds.
2. This work is technical solid. It proves in detail the theory that the MSG is a neural ODE solver.
3. The proposed MSG outperforms existing spiking GNNs in both node classification and link prediction tasks with low energy consumption.

**Weaknesses:**

1. Figure 4(a) is confusing. In line 356-357, the author state that backward time of DvM is significantly less than that of BPTT algorithm. But in figure 4(a), the bar of DvM is higher. Is this a mistake? If not, please explain. In addition, it would be helpful to list the complete training times.
2. Figure 2 shows that the manifold representations will feed forward with the spike trains. This is acceptable during training, but during inference, since $f(\cdot)$ is an exponential function, the computation of the manifold representations involves a lot of float-point operations, which is incompatible with the nature of spiking neural networks. Please describe in detail the computation of the manifold representations during inference, especially the float-point operations involved.

**Questions:**

Please refer to the weaknesses

**Limitations:**

The authors have stated the limitations and broader impact.

---

> ### Author Rebuttal · Authors · 2024-08-05
>
> **W1: On Figure 4(a) and Training Time**
>
> Sorry for the typo in Figure 4(a) where we mislabeled the legend. We list the training time as follows.
>
> The gradient backpropagation (BP) time of DvM and Surrogate on Lorentz model is shown below.
>
> |  Time steps | *BP Times (DvM)*  | BP Times (Surrogate) |
> |--- | ----|--- |
> | 30 | 0.0405|  0.0523|
> | 50 | 0.0408 | 0.1054 |
> | 70 | 0.0418|  0.6098|
> | 100 | 0.0621|  2.3260|
>
> The gradient backpropagation (BP) time on Sphere model is as follows.
>
> |  Time steps | *BP Times (DvM)*  | BP Times (Surrogate) |
> |--- | ----|--- |
> | 30 | 0.0075|  0.0224|
> | 50 | 0.0076 | 0.0605 |
> | 70 | 0.0082|  0.1572|
> | 100 | 0.0333|  1.4022|.
>
> As shown in the tables, the proposed DvM achieves the significant less training time to BPTT, as DvM does not perform gradient backpropagation recursively for each time step.
>
> **W2: Details of model inference.**
>
> In the model inference, **the proposed MSG do not need the expensive successive exponential maps, and thereby only limited float-point operations (i.e., addition) are involved.**
>
> In MSG, we leverage the tangent vectors $\mathbf{v}$’s for the downstream tasks, in accordance to the Charts of manifold ODE in Sec. 5.
>
> Specifically, with the model parameters trained by DvM,
> we first conduct the pooling operation over the spikes to obtain the tangent vector as in Equation 7,
>
> and then sum over each layer to get $\mathbf{v}_i=\epsilon\sum_l \mathbf{v}^{l-1}_i$.
>
> Finally, $\mathbf{v}_i $ is feed in the classification head; or the Fermi-Dirac decoder for link prediction.
>
> We will specify this point in the final version.

---

> > ### Comment · Reviewer_RDnD · 2024-08-13
> >
> > Thanks for your response. My concerns have been addressed.

---

### Official Review · Reviewer_aPZg · 2024-07-11

**Soundness:** 3
**Presentation:** 3
**Contribution:** 3
**Rating:** 6
**Confidence:** 3

**Summary:**

In order to improve energy efficiency and performance in graph learning, the research presents a unique Manifold-valued Spiking Graph Neural Network (MSG), which combines spiking neurons with Riemannian manifolds. Differentiation via Manifold (DvM), a unique training approach, is proposed by the authors to solve the high latency issue with spiking GNNs.

**Strengths:**

Combining spiking neurons with Riemannian manifolds is a novel idea that addresses both energy efficiency and performance

Numerous experiments show that MSG is more effective and energy-efficient than both traditional and spiking GNNs currently in use.

**Weaknesses:**

It is not the first time the Riemann manifold has been introduced into spiking neural networks. The authors lack proper discussion of these works.

The energy consumption is unknown for the proposed method due to the complexity of integrating spiking neurons with Riemannian manifolds.


While the theoretical contributions are significant, the paper lacks discussion on practical applications and real-world scenarios.

**Questions:**

Can the authors provide more details on the potential neuromorphic implementation?

Are there any specific real-world applications where MSG could be particularly beneficial?

**Limitations:**

See weakness and questions

---

> ### Author Rebuttal · Authors · 2024-08-05
>
> **W1: It is not the first time the Riemann manifold has been introduced into spiking neural networks.**
>
> In this paper, the geometric concept of Riemannian manifold is **a smooth manifold endowed with a Riemannian metric**. To the best of our knowledge, this is the first time that Riemannian manifold is introduced into spiking neural networks.
> Could you please provide the relevant paper?
>
> **W2: The energy consumption is unknown for the proposed method.**
>
> The result of theoretical energy consumption is given in Table 3 where both encoding process energy and spiking process energy are calculated.
>
> Specifically, we calculate the number of multiply-and-accumulate operations (MACs), and the number of synaptic operations operations (SOPs), following the previous works [1] [2]. The formula of theoretical energy consumption is given below,
> $$
> E=E_{\text{encoding}}+E_{\text{spiking}}=E_{\mathrm{MAC}}\sum_{t=1}^{T}Nd S_t+E_{\mathrm{SOP}}\sum_{t=1}^{T}\sum_{l=1}^{L}S_{t}^{l},
> $$
> - \\( N\\): the number of nodes in the graph
> - \\( d\\): the dimension of node features
> - \\( L\\): the number of layers in the neural model.
> - \\( T\\): the time steps of the spikes
> - \\( S_l^t\\): the output spikes at time step $t$ and layer $l$.
>
> More details are given in Appendix E.3.
>
> In addition, **in the model inference, the proposed MSG leverages the tangent vectors for the downstream tasks (i.e., we sum the $\epsilon\mathbf{v}^{l-1}_i$ in Eq. (7) accumulatively for each layer.), and do not need the expensive exponential maps in the manifold**. Consequently, the proposed MSG achieves low energy consumption.
>
> [1] Li, J., H. Zhang, R. Wu, et al. A Graph is Worth 1-bit Spikes: When Graph Contrastive Learning Meets Spiking Neural Networks. arXiv preprint arXiv:2305.19306, 2023.
>
> [2] Zhu, Z., J. Peng, J. Li, et al. Spiking graph convolutional networks. In Proceedings of the 31st IJCAI, pages 2434–2440. ijcai.org, 2022.
>
> **W3/Q2: On the real-world applications where MSG could be particularly beneficial.**
>
> First, the proposed MSG is particularly beneficial to the **Large-scale Riemannian Graph Learning**.
>
> Riemannian graph models (e.g., HGCN of hyperbolic space, $\kappa$-GCN of constant curvature manifold) report superior performance in recent years. Fundamentally, Riemannian models are well aligned with the graph structures, but they are computationally expensive and thus results in high energy consumption.
>
> Bridging this gap, MSG essentially decreases the energy consumption of Riemannian GNN, which is shown in Table 3. We list the theoretical energy consumption (mJ) of HGCN, $\kappa$-GCN and the proposed MSG here for reference.
>
> | Model | Computers | Photo | CS | Physics |
> | --- | --- | --- | --- | --- |
> | HGCN | 1.614 |0.869 | 18.390 | 42.800 |
> | $\kappa$-GCN | 1.647 | 0.889 | 18.440 | 42.836|
> | MSG | 0.047 | 0.043 | 0.026 | 0.029|
>
> Consequently, the proposed model is much more scalable and greener to handle large graphs.
>
> Second, the proposed MSG is particularly beneficial to the **SNN-based Link Prediction.**
>
> Existing SNNs do not achieve satisfactory results in link prediction, but MSG achieves competitive link prediction results to the state-of-the-art ANNs with the aid of Riemannian manifold, as shown in Table 1.
>
> **Q1: Details on the potential neuromorphic implementation.**
>
> The neuromorphic implementation of the proposed MSG has **no difference to the previous Euclidean Spiking GNNs**, such as SpikeGCL and SpikeGCN.
>
> Specifically, referring to the model inference (the answer of W2), the proposed MSG only needs simple operations besides SNN, and do not need the complicated manifold operations, which is similar to the previous methods, such as SpikeGCL and SpikeGCN. Also, the spikes can be directly implemented on neuromorphic hardware [1][2].
>
> [1] J. Zhao, E. Donati and G. Indiveri, "Neuromorphic Implementation of Spiking Relational Neural Network for Motor Control," 2020 2nd IEEE International Conference on Artificial Intelligence Circuits and Systems (AICAS), Genova, Italy, 2020, pp. 89-93, doi: 10.1109/AICAS48895.2020.9073829.
>
> [2] Thiele J C, Bichler O, Dupret A. Spikegrad: An ann-equivalent computation model for implementing backpropagation with spikes[J]. arXiv preprint arXiv:1906.00851, 2019.

---

### Official Review · Reviewer_NhQj · 2024-07-12

**Soundness:** 4
**Presentation:** 3
**Contribution:** 4
**Rating:** 8
**Confidence:** 5

**Summary:**

This paper first analyzes the limitations of spiking GNN, representation space and model training, and then present a Riemannian model called MSG, which is connected to manifold ODE. Finally, the authors conduct experiment to show the effectiveness and energy efficiency.

**Strengths:**

On the significance,
1.This paper relates two previously disparate research areas, SNN and Riemannian geometry.
2.Energy efficiency is of importance to the wide use of Riemannian models, given that they are typically computationally expensive.

On the originality,
1.Unlike existing Riemannian GNNs, the authors design a Riemannian GNN with the spiking neuron.
2.It is a novel idea to parallel Riemannian and spiking domains, providing the opportunity for a faster training algorithm, DvM.

On the quality, this is a Solid paper with few technical,
1.The theoretical results (connection to ODE) are proved, and the closed form gradient is well elaborated.
2.The empirical results are adequate, and Codes are provided for reproducibility.

On the clarity, this paper is well-organized and easy to follow in general.

**Weaknesses:**

1.In the experiment, in Table 1, the authors do not provide some of link perdition results of SNNs. Thus, it needs to specify on those SNN baselines, e.g., is SpikeNet [33] able to conduct link prediction? Why not?
2.This paper requires systematic understanding of differential geometry, and thus is not friendly to the reader not familiar to this area.

**Questions:**

See W1, and other questions are listed as follows:
1.I have double checked Proofs 4.1 and 5.1, and the proofs are correct. However, I would like to clarify: whether or not there is a typo in Line 275 of Thm 5.2, that z and y are swapped.
2.I notice that, the authors mention another optimization method, sub-gradient [15], also different from the BPTT scheme. Thus, specify the sub-gradient, and the different between sub-gradient and the proposed DvM.
3.Given that gradient exploding and vanishing are common problems in training SNN, will MSG suffer from these issues? Also, is there any treatment for the numerical stability of Riemannian operators, e.g, sinh and cosh in the Appendix?

**Limitations:**

No limitation (negative societal impact) needs to be specifically addressed.

---

> ### Author Rebuttal · Authors · 2024-08-05
>
> **W1: In Table 1, some link prediction results of SNN baselines are lacked.**
>
> For the SNNs generating spiking representations (i.e., SpikeGCN and SpikeGCL), we feed the spiking representation into Fermi-Dirac decoder, obtaining the pairwise similarity for link prediction.
>
> For the SNNs that cannot generate spiking representations (i.e., SpikeNet and SpikeGraphormer), we do not report the results of link prediction, since there exists no well-recognized similarity for the spikes.
> Also, SpikeNet is designed for node classification specifically.
>
>
> **Q1: Typo in Line 275.**
>
> Thanks for your careful review and kind reminder. Concretely, $\mathbf{y(t)}=Log_\mathbf{z}(\mathbf{z}(t))$ in the Theorem.
>
> **Q2: The difference between the sub-gradient method in reference [15] and the proposed DvM approach.**
>
> The sub-gradient method generates spike representations from corresponding spike trains, and conduct gradient backpropagation through the spike representations.
>
> The sub-gradient method works with the Euclidean space of spike representations, while the proposed DvM conducts gradient backpropagation on the manifold.
>
> In the sub-gradient method, the clamp operator may cause gradient error, while each term in DvM is differentiable, and thus DvM computes gradient without approximation.
>
> **Q3: Will proposed MSG suffer from gradient exploding/vanishing?**
>
> SNNs are exposed to the issue of gradient exploding/vanishing due to BPTT training, which requires recursive gradient backpropagation for each timestep. Thus, SNN tends to encounter gradient exploding/vanishing when the number of time steps is large (similar to the RNN going deeper).
>
> In contrast, in MSG, we propose DvM training approach that conducts gradient backpropagation via the manifold and does NOT need recursive backpropagation. Moreover, from Eq. (39) in Appendix C, our formulation with the exponential map is similar to the residual connection, alleviating the gradient vanishing.
>
> Empirically, as shown in Figure 4(a) and the results in Appendix F, the proposed MSG does not expose to gradient exploding/vanishing.
>
> **On the numerical stability of cosh and sinh.**
> We address the issue of numerical stability in the division of $cosh$ and $sinh$, especially for the zero-value.
>
> Specifically, we compute the limits in the form of $\frac{sinh(x)}{x}$ and $\frac{cosh(x)-1}{x^2}$ when $x\rightarrow 0$. We substitute the limits into the formulas as $x$ is less than a threshold.
> $$
> \lim_{x\rightarrow 0}\frac{sinh(x)}{x} = 1.
> $$
>
> $$
> \lim_{x\rightarrow 0}\frac{cosh(x)-1}{x^2} = 0.5.
> $$

---

> ### Comment · Reviewer_NhQj · 2024-08-08
>
> Thanks for the authors' response, and the responses have addressed my concerns. Having reviewed the comments from other reviewers, I have decided to endorse this paper, and raise my score to Strong Accept.

---

### Official Review · Reviewer_BtYr · 2024-07-16

**Soundness:** 4
**Presentation:** 4
**Contribution:** 4
**Rating:** 8
**Confidence:** 4

**Summary:**

This paper focuses on the theoretical aspects of spiking neural networks, which are a variant of neural networks closer to the human brain, particularly incorporating a time component. One of the hopes for this type of networks is that they are much more energy efficient. In this paper, the authors extend the classical spiking graph neural networks setting, which only handle graphs in Euclidean space, to the setting of Riemannian manifolds. This is highly non-trivial, requiring a specific of each spiking neuron on manifolds. Intriguingly, they then relate this new model to manifold ODEs, and also show even theoretically a significant improvement of energy efficiency as opposed to classical GNNs.

**Strengths:**

* SNNs are very timely due to the severe energy problems of classical NNs.
* Goes significantly beyond the state of the art.
* Very well-organized paper, intriguing to read.
* From deep math to numerical experiments, this paper covers the full range
* Comprehensive analysis of the newly introduced type of SNNs, and also showing their superiority

**Weaknesses:**

* Argument, why Riemannian manifolds are required, could be strengthened, and also what about other types of manifolds?

**Questions:**

see weaknesses

**Limitations:**

---

---

> ### Author Rebuttal · Authors · 2024-08-05
>
> **Q: Argument, why Riemannian manifolds are required, could be strengthened, and also what about other types of manifolds?**
>
> On the one hand, Riemannian manifolds are well aligned with graph structures (e.g., the hierarchical/tree-like and cyclical structures of graphs correspond to hyperbolic and spherical manifold, respectively), and Riemannian graph models (e.g., HGCN of hyperbolic space [1], $\kappa$-GCN of constant curvature manifold [2]) report superior performance in recent years.
>
> Note that, besides matching the graph geometry, the proposed MSG essentially decreases the energy consumption of Riemannian GNN, which is shown in Table 3. **Thus, the proposed model is much more scalable and greener to handle large graphs.**
>
> On the other hand, owing to the Diffeomorphism of Riemannian manifolds, the proposed MSG obtains manifold representation based on SNN. **As a result, MSG becomes a more general approach that can handle not only node classification but also link prediction.**
>
> Note that, existing SNNs do not achieve satisfactory results in link prediction, but MSG achieves competitive link prediction results to the state-of-the-art ANNs with the aid of the Riemannian manifold.
>
> **For other types of manifolds:**
>
> The diffeomorphism and distance measure of Riemannian manifold essentially support the design of DvM optimization approach and benefit link prediction, respectively.
>
> For pseudo-Riemannian manifolds, we can apply the idea of MSG but it lacks an elegant exponential map, given the geodesic is broken in pseudo-Riemannian manifolds [3]. Thus, we consider the Riemannian manifolds, e.g., hyperbolic space, sphere, and their products.
>
> For the topological manifolds other than Riemannian/pseudo-Riemannian ones, the alignment between graph and the manifold is not well established.
>
> [1] Chami, I., Ying, Z., Re, C., and Leskovec, J. Hyperbolic graph convolutional neural networks. In Advances in the 32nd NeurIPS, pp. 4869–4880, 2019.
>
> [2] Bachmann, G., G. Bécigneul, O. Ganea. Constant curvature graph convolutional networks. In Proceedings of the 37th ICML, vol. 119, pages 486–496. PMLR, 2020.
>
> [3] Xiong, B., S. Zhu, N. Potyka, et al. Pseudo-riemannian graph convolutional networks. In Advances in NeurIPS, vol. 32. 2022.

---

### Author Rebuttal · Authors · 2024-08-07

Thanks for the appreciation and detailed comments of all the reviewers! Tables in the rebuttal are also given in the PDF for better readability.



All authors of submission 6586.

---

### Decision · Program_Chairs · 2024-09-25

**Decision:**

Accept (poster)

**Comment:**

I have read all comments and responses. Reviews appear to be consentaneous, with five scores of 8, 8, 8, 6, and 6. All concerns of reviewers have been fixed. It is recommended to accept this manuscript.